# ANXA11 biomolecular condensates facilitate protein-lipid phase coupling on lysosomal membranes

Jonathon Nixon-Abell [1] ✉, Francesco S. Ruggeri [2,3], Seema Qamar[1], Therese W. Herling [3], Magdalena A. Czekalska [3], Yi Shen [3,4,5], Guozhen Wang[1,12], Christopher King[6], Michael S. Fernandopulle[1,6,7], Tomas Sneideris[3], Joseph L. Watson [8], Visakh V. S. Pillai [2], William Meadows[1], James W. Henderson[1], Joseph E. Chambers [1], Jane L. Wagstaff[9], Sioned H. Williams[1], Helena Coyle[1], Greta Šneideriené[3], Yuqian Lu[3], Shuyuan Zhang[3], Stefan J. Marciniak [1], Stefan M. V. Freund [9], Emmanuel Derivery [8], Michael E. Ward [6], Michele Vendruscolo [3], Tuomas P. J. Knowles [3] & Peter St George-Hyslop [10,11] ✉

Phase transitions of cellular proteins and lipids play a key role in governing the organisation and coordination of intracellular biology. Recent work has raised the intriguing prospect that phase transitions in proteins and lipids can be co-regulated. Here we investigate this possibility in the ribonucleoprotein (RNP) granule-ANXA11-lysosome ensemble, where ANXA11 tethers RNP granules to lysosomal membranes to enable their co-trafficking. We show that changes to the protein phase state within this system, driven by the low complexity ANXA11 N-terminus, induces a coupled phase state change in the lipids of the underlying membrane. We identify the ANXA11 interacting proteins ALG2 and CALC as potent regulators of ANXA11-based phase coupling and demonstrate their influence on the nanomechanical properties of the ANXA11-lysosome ensemble and its capacity to engage RNP granules. The phenomenon of protein-lipid phase coupling we observe within this system serves as a potential regulatory mechanism in RNA trafficking and offers an important template to understand other examples across the cell whereby biomolecular condensates closely juxtapose organellar membranes.

Intracellular phase transitions of both protein and lipid components have emerged as key physical principles affecting the material properties and organisation of various biological systems. Moreover, across eukaryotic cells there are now numerous examples of protein phase transitions occurring in close proximity to lipid membranes. This raises the intriguing possibility that phase transitions in proteins and lipids could be functionally coupled.

Phase transitions of cellular lipids have been an intense area of study dating back several decades[1–4]. The field has primarily focussed on the transition between disordered and ordered lipid phase states. In the disordered phase, lipids are highly fluid and are free to laterally diffuse within the membrane, whereas in the ordered phase, lipids become less mobile and more tightly packed. These changes in lipid mobility and crowding have been shown to profoundly impact the mechanical stiffness and nano-domain architecture of biological membranes[5]. Such effects work in conjunction with protein-protein and protein-lipid interactions between structured proteins on, or within, the membrane to

shape diverse biological processes, ranging from transmembrane signal transduction to viral budding[6,7].

Recent work has also uncovered an important role for protein phase transitions within cells, in particular, during the formation of biomolecular condensates. Biomolecular condensates (henceforth condensates) are heteromeric assemblies of proteins and/or nucleic acids that form intracellular membraneless compartments. Their assembly occurs as constituent proteins/nucleic acids undergo phase transitions from a dispersed/dilute phase to a condensed phase. Numerous functionally distinct condensates have now been identified, playing important roles in diverse cellular processes ranging from transcription to intracellular signalling.

A subset of these condensates forms on, or near, cellular membranes. Specific examples include: the T-cell receptor (TCR); the post-synaptic density 95 (PSD95) complex; synapsin-associated presynaptic vesicles; TIS granules and Sec bodies on the surface of the ER; P bodies on mitochondria; and recently the ribonucleoprotein (RNP) granule-ANXA11-lysosome complex. These membrane-associated condensates are involved in an assortment of cellular functions spanning trans-membrane signalling, intracellular vesicle dynamics, and cargo trafficking[8–16]. The juxtapositioning of phase transitioning proteins and membrane lipids within these systems suggests that they could modulate each other's phase state. For example, a lipid phase transition might influence the local accumulation, clustering, and phase state of protein components in adjoining biomolecular condensates. Conversely, protein components directly apposed to membranes could alter the local composition, ordering and phase state of subjacent lipids. Indeed, theoretical and experimental studies have demonstrated that such a phenomenon likely plays an important role at the plasma membrane and, specifically, in the organisation of immune synapses[17–23].

Here, we investigate phase coupling within a functional ensemble involving a cytoplasmic condensate (RNP Granule) and a membrane-bound intracellular organelle (the lysosome). We have shown previously that the components of this ensemble are tethered together by ANXA11 to enable the long distance co-trafficking of RNP granules with lysosomes[16]. ANXA11 engages the RNP granule via a low complexity N-terminal domain. At its C-terminus, ANXA11 contains a structured annexin repeat domain which binds to negatively charged phospholipid headgroups in the lysosomal membrane in a calcium-dependent manner.

We demonstrate that ANXA11 binding to membranes induces coupled changes in the phase state of the subjacent lipids, as has been reported for other annexin family members[24–26]. However, we show that the magnitude of this lipid phase transition can be tuned according to the phase state of the ANXA11 low complexity domain. We next demonstrate that two known protein interactors of ANXA11, apoptosis-linked gene 2 (ALG2; *PDCD6*[27]) and calcyclin (CALC; *S100A6*[28]), modulate the phase state of the ANXA11 low complexity domain. We observe that ALG2- and CALC-mediated changes in the ANXA11 phase state evoke a coupled phase state change in the underlying lysosomal lipid membrane. A functional consequence of this phase coupling is the modulation of the mechanical properties of the ANXA11-lysosome ensemble, which correlates with its capacity to engage RNP granules. Together, these findings support the notion that the interaction of biomolecular condensates with intracellular membrane bound organelles can result in coupled changes in their biophysical properties and functions. As such, these observations provide a framework for understanding how lipid and protein phase coupling may occur in other organellar membrane-associated condensates across the cell.

## Results

### The ANXA11 LCD drives a dispersed to condensed ANXA11 phase transition, while the ARD facilitates lipid membrane binding

To explore the possibility of phase coupling in the RNP granule-ANXA11-lysosome ensemble, we built an in vitro reconstitution model comprising recombinantly expressed ANXA11 and giant unilamellar vesicles (GUVs) containing the lysosomally-enriched phospholipid phosphatidylinositol 3-phosphate (PI(3)P), which we have shown previously serves as the principal binding target for ANXA11[16].

The full length ANXA11 protein (aa 1-502; henceforth **FL**) comprises a mostly disordered N-terminal low complexity domain (aa 1-185; henceforth **LCD**), and a structured C-terminal annexin repeat domain (aa 186–502; henceforth **ARD**) (Fig. 1a, Supplementary Fig. 1A, B). First, to explore the contributions of the LCD and ARD in condensate formation in solution, we incubated fluorescently labelled recombinant FL, LCD or ARD (Supplementary Fig. 4A) at physiological pH and temperature conditions. Supporting our previous observations[16], the LCD is both necessary and sufficient for inducing an ANXA11 phase transition from a dispersed to a condensed state (Fig. 1b, Supplementary Movie 1). Importantly, this transition exhibits several classical hallmarks of a liquid-liquid phase transition. It is concentration dependent (Fig. 1b), the condensates are spherical (Fig. 1b), and can be dissolved and reformed with temperature cycling assays (Supplementary Fig. 1c, d). Also, FTIR spectra of the Amide I absorption region reveal only minor differences (<5%) in intermolecular β-sheet content between the dispersed and condensed state, indicating an absence of protein aggregation (Supplementary Fig. 1E)[29,30].

We next investigated the roles of the LCD and ARD hemiproteins in driving ANXA11-membrane binding. As we have shown previously, ANXA11 binds to PI(3)P-containing GUV membranes in a $Ca^{2+}$-dependent manner (Supplementary Fig. 1F, G)[16]. Here we demonstrate, across a range of $Ca^{2+}$ concentrations, that this binding is entirely dependent on the ARD. The LCD hemiprotein alone exhibits negligible recruitment to GUV membranes (Fig. 1c, d).

### ANXA11-lipid binding causes a liquid-to-gel phase transition of lipid membranes

The binding of several other annexin family members has been shown to alter the organisation and phase state of lipid membranes[24,25]. To interrogate this possibility with ANXA11, we first examined the mobility of the fluorescently-labelled PI(3)P in our GUVs using fluorescence recovery after photobleaching (FRAP) (Fig. 2a). Binding of ANXA11 FL to GUVs caused a marked reduction in PI(3)P mobility (Fig. 2b). We also observed a similar reduction in the mobility of dioleoylphosphatidylethanolamine (DOPE) – another lipid within our GUVs (Supplementary Fig. 2A). By incubating GUVs with the ARD hemiprotein, we determined that these changes in lipid diffusivity are entirely attributable to ARD-lipid interactions. The LCD of ANXA11 does not bind membranes (Fig. 1c, d) and so was not assessed.

To elucidate the basis of ANXA11's effect on lipid mobility, we utilised a solvatochromic dye (PK dye) that reports on membrane lipid phase state. The PK dye is a lipophilic push pull pyrene probe that intercalates into membranes and shifts its fluorescence emission based upon the local lipid order/disorder[31]. The ratio between a blue/green (449–550 nm) and red (550–650 nm) emission window thus reports the relative lipid order, or phase state, of the membrane (Fig. 2c). Initially, to validate the PK dye in our system, we warmed our GUVs from 18 to 37 ℃, where we observed an expected temperature-dependent decrease in lipid order (Supplementary Fig. 2B)[32,33]. We thus reasoned that we could use the PK dye to assess whether ANXA11's effects on lipid mobility were underpinned by a phase transition in GUV membranes. Indeed, the $Ca^{2+}$-dependent binding of either ANXA11 FL or the ARD hemiprotein to GUVs both induced a significant, and comparable, increase in lipid ordering (Fig. 2d, e). Of note, the composition of our GUVs is such that one would not predict phase *separation* of the lipids into distinct ordered and disordered domains[34]. Instead, our findings indicate a bulk phase transition of the lipid membrane into a more ordered state.

To determine the magnitude of this lipid phase transition, we employed atomic force microscopy-based infrared (IR) nanospectroscopy

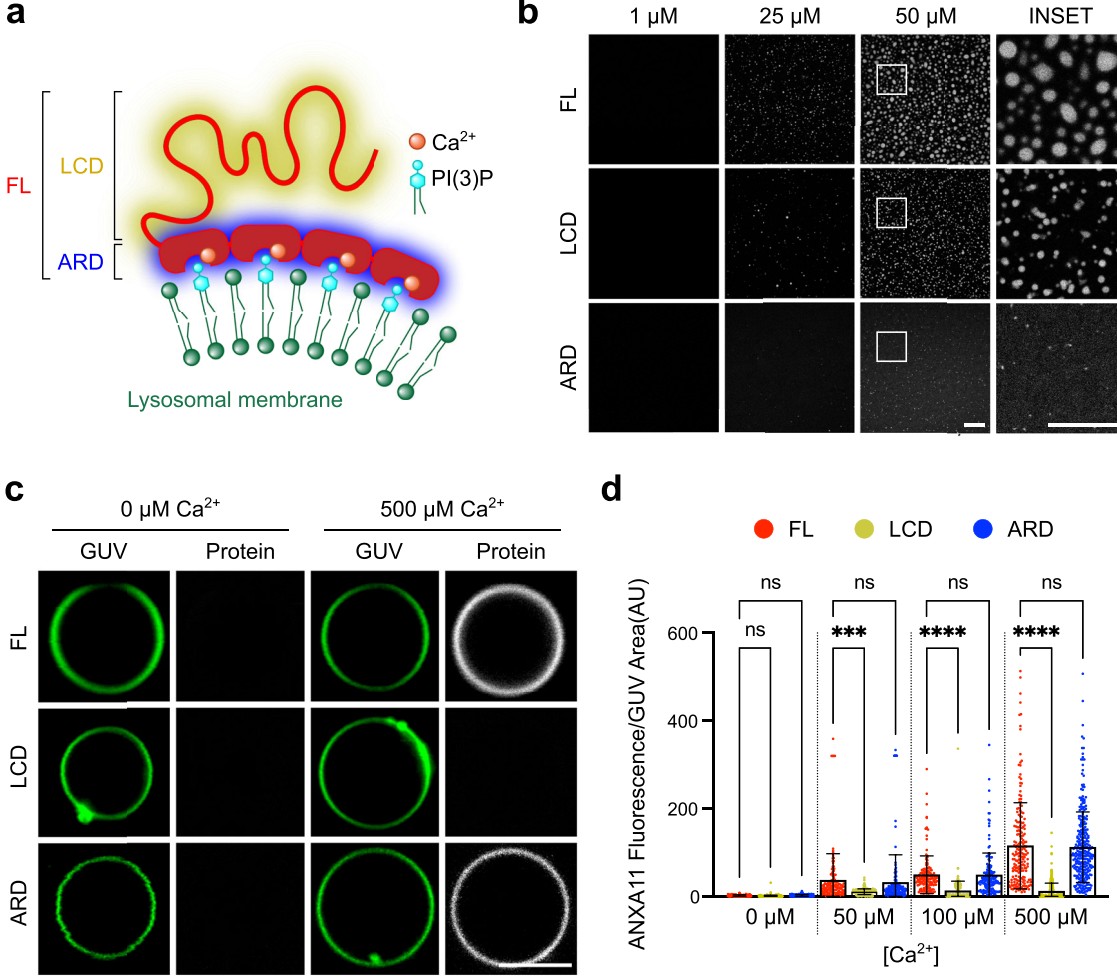

**Fig. 1 | ANXA11 is a structurally bipartite protein capable of biomolecular condensation and lipid binding. a** A schematic of full length (FL) ANXA11 binding to lysosomal membranes, illustrating the $Ca^{2+}$-dependent annexin repeat domain (ARD) association with PI(3)P, and the cytosolic-facing low complexity domain (LCD). **b** Fluorescence micrographs of recombinant AF647-labelled FL (aa 1-502), LCD (aa 1-185) and ARD (aa 186–502) ANXA11 at varying protein concentrations. Scale bar −5 μm. Representative images repeated in three independent experiments

(**c**) Representative fluorescence images of ATTO488 GUVs incubated with 0.5 μM AF555-labelled ANXA11 FL, LCD and ARD in the presence or absence of 500 μM $Ca^{2+}$. Scale bar −5 μm. **d** Quantification of the fluorescence intensity of AF555-labelled FL, LCD and ARD recruited to GUVs as shown in (**c**) at varying $Ca^{2+}$ concentrations. Mean ± SD. Kruskal-Wallis test with Dunn's multiple comparison, ***$p$ = 0.0002, ****$p$ < 0.0001, ns - not significant ($p$ > 0.05), $n$ = 3 repeats (110-379 GUVs).

(AFM-IR)[35,36]. AFM-IR exploits a ~ 20 nm diameter probe to simultaneously extract nanoscale morphological and chemical properties of protein and lipid biomolecular systems (Fig. 2f, see Infrared Nanospectroscopy Methods section). To perform the AFM-IR spectroscopy, we deposited GUVs coated in either ANXA11 FL or ARD hemiprotein onto hydrophobic ZnSe surfaces (Fig. 2g, h, Supplementary Fig. 2C). From this we were able to extract distinct protein and lipid IR absorption spectra (Fig. 2i, Supplementary Fig. 2D-G). Spectra from GUVs without bound ANXA11 possessed typical C=O peaks for phospholipids and cholesterol esters at 1730−32 cm⁻¹ (green line, Fig. 2j). This is consistent with the lipids existing in a liquid phase, based on previous studies on lipids of a similar composition[33]. However, binding of either ANXA11 FL or ARD caused a statistically significant 8 cm⁻¹ increase in wavenumber in the position of the lipid C=O peak (Fig. 2j, k). The magnitude of this shift indicates a phase transition of the lipids into a gel-like state[37,38]. Of note, analysis of the protein IR spectra (Supplementary Fig. 2H) reveals that ANXA11 does not exhibit significant secondary structural changes upon GUV binding (Supplementary Fig. 2I, J). This result negates the possibility that the lipid phase state change is a result of aggregation of ANXA11 on the membrane.

Taken together, these data indicate that ANXA11 binding to lysosomal-like GUV membranes causes a liquid-to-gel phase transition in the underlying lipids. This effect appears to be entirely mediated by

the structured ARD. However, we speculated that modulation of the phase state of ANXA11, through its LCD, might influence the magnitude of this observed lipid phase transition.

## ANXA11 condensation mediates coupled changes in the phase state of lipid membranes

To interrogate the possibility of coupled protein-lipid phase state changes, we sought to induce a phase transition in membrane-bound ANXA11 into a condensed state to observe how this affected the underlying lipid phase state.

Initially, we investigated whether membrane-bound ANXA11 could indeed undergo condensation. Typically, the phase state of protein components in in vitro systems can be modulated by changing the physical or chemical conditions (e.g., [salt], temperature, macromolecular crowding). However, altering these parameters also has a direct effect on the lipid phase state and/or integrity of GUV membranes. To avoid this confound, we exploited the specialised domain architecture of ANXA11. We reasoned that membrane binding by ANXA11 ARD, which cannot undergo condensation (Fig. 1b), should be driven by simple protein-lipid interactions. However, if ANXA11 FL can undergo condensation on membrane surfaces, then its recruitment to GUVs will be a function of both ARD-mediated protein-lipid

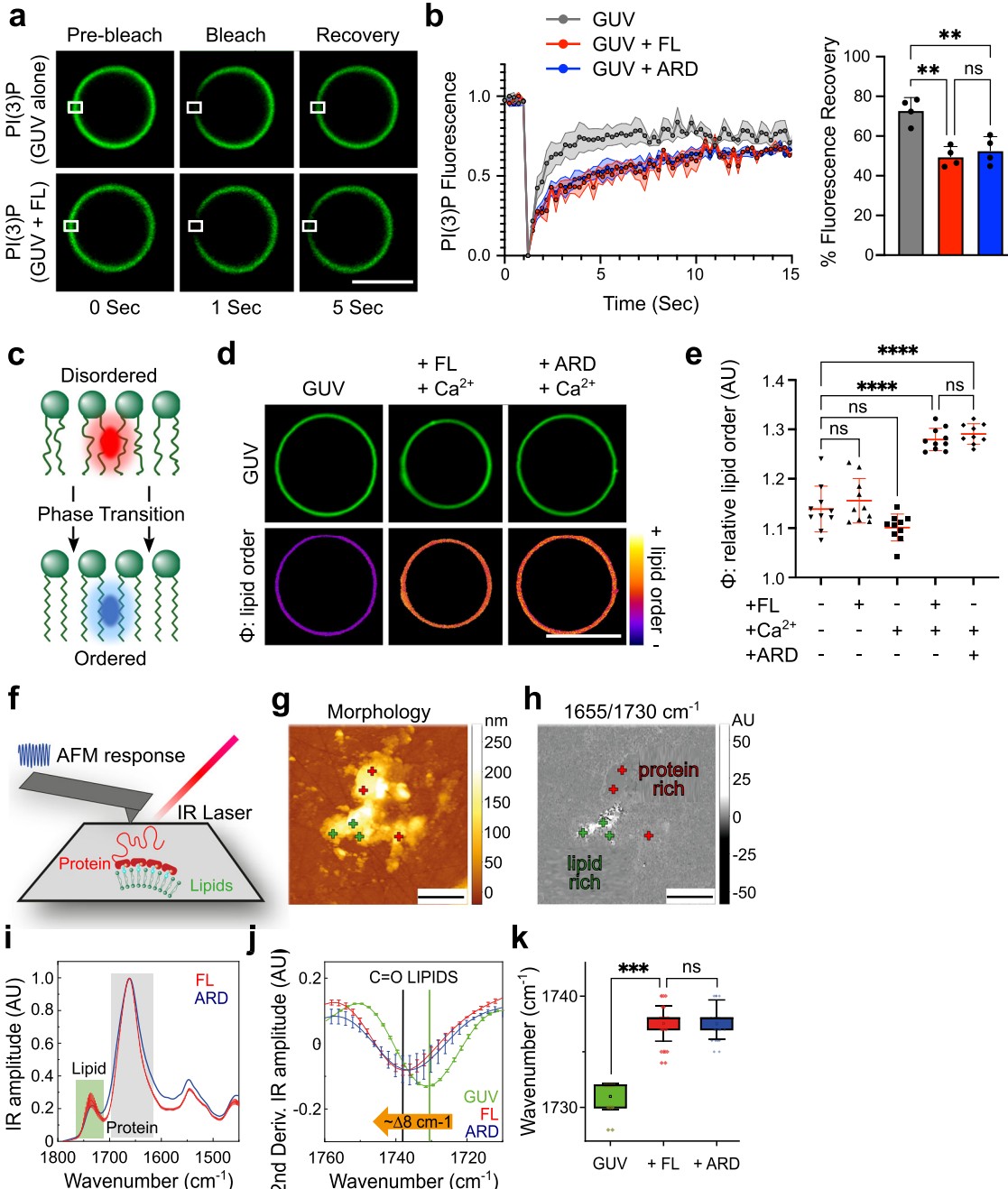

**Fig. 2 | ANXA11-GUV binding causes a liquid-to-gel phase transition of lipid membranes. a** Lipid FRAP series of BODIPY-PI(3)P in GUVs in the presence of 100 μM Ca²⁺ with and without 0.5 μM ANXA11 FL. Scale Bar-5 μm. **b** PI(3)P fluorescence recovery rates in the presence of 100 μM Ca²⁺ co-incubated with either 0.5 μM ANXA11 FL or 0.5 μM ARD. The % fluorescence recovery by 5 s is plotted alongside. Mean ± SD. One-way ANOVA with Tukey's multiple comparison, **$p = 0.0014$(Ca vs FL) ;0.0036(Ca vs ARD), ns−not significant ($p > 0.05$), $n = 4$ GUVs. **c** Schematic illustrating the change in PK dye fluorescence emission as lipids transition from a disordered (red emission) to ordered (blue emission) state. **d** Fluorescence images of ATTO647 GUVs labelled with 5 μM PK dye to extract the relative lipid order (φ) of GUVs alone or with 100 μM Ca²⁺ and either 0.5 μM ANXA11 FL or 0.5 μM ARD. Scale Bar-5 μm. **e** Quantification of data displayed in (**d**), including controls. Mean ± SD. One-way ANOVA with Tukey's multiple comparison, ****$p < 0.0001$, ns−not significant, $n = 3$ repeats (30−82 GUVs). **f** Schematic of AFM-IR setup which acquires

nanoscale resolved chemical spectra of protein and lipid components. **g** AFM-IR morphology maps of GUV fragments bound to ANXA11. Green crosses = 'lipid only' signature. Red crosses = 'protein and lipid' signatures. Scale bars-2 μm. **h** Ratio of the infrared maps from protein:lipid (1655/1730 cm⁻¹) spectroscopic signatures. Scale bars−2 μm. **i** Comparison of the AFM-IR spectra of GUV fragments bound to 0.5 μM ANXA11 FL and 0.5 μM ARD at 100 μM Ca²⁺. Mean ± SD $n = 5$ (12−31 GUV fragments) (**j**) Second derivatives of the spectra in the IR absorption of the C = O stretching region of lipids for GUV fragments alone or bound to either ANXA11 FL or ARD. Mean ± SD, $n = 5$ (12−31 GUV fragments) (**k**) Quantification of the wavenumber of GUV fragments alone compared with those bound to ANXA11 FL or ARD. Mean ± SD and 25−75th percentile (box). One-way ANOVA with Tukey's multiple comparison, ***$p = 0.0023$, ns−not significant ($p > 0.05$), $n = 5$ repeats (12−31 GUV fragments).

interactions *and* LCD-based protein-protein condensation. By exploring ANXA11 recruitment to GUVs across a range of protein concentrations, we demonstrated that ANXA11-lipid interactions (as mediated through the ARD) are saturated at ~10 μM. However, collective interactions driven by ANXA11-LCD domains, fail to reach saturation even at ~30 μM (Fig. 3a). This indicates that ANXA11 recruitment to membranes is mediated at low concentrations by ARD-lipid interactions, then upon saturation of lipid binding sites, through collective LCD-LCD based interactions between independent ANXA11 molecules. This collective binding is characteristic of interactions underpinning protein condensation[39].

To confirm that this collective binding profile occurred as a result of LCD-based ANXA11 condensation, we reasoned that adding LCD hemiprotein to FL ANXA11 should promote ANXA11 condensation. Indeed, exogenously added LCD hemiprotein co-condenses with FL protein (Supplementary Fig. 3A). This causes a shift in the phase boundary of FL ANXA11 in favour of the condensed state (Supplementary Fig. 3B), resulting in larger and more numerous condensates (Fig. 3b). This is not simply a protein crowding effect, as a purified Halo protein control fails to incorporate into FL ANXA11 condensates (Supplementary Fig. 3A) and does not impact the phase boundary (Supplementary Fig. 3B) or size/number of ANXA11 condensates (Figure 3A). Importantly, FRAP studies revealed that the addition of LCD hemiprotein also pushes the ANXA11 condensates into a more immobile state (Supplementary Fig. 3C). These changes in mobility likely represent a shift of the condensates into a more gel-like state because they do not exhibit morphological or secondary structural changes found in protein aggregates (Supplementary Fig. 3D, E). Once again, addition of the Halo control had no effect on ANXA11 mobility (Supplementary Fig. 3C). Collectively, these data suggest that the addition of exogenous LCD hemiprotein to FL ANXA11 promotes the dispersed to condensed phase transition of the full-length protein and pushes the ANXA11 into a more gel-like state.

To test whether such effects could also occur on lipid surfaces, we took GUVs coated in either ANXA11 FL or ARD hemiprotein and added exogenous fluorescently labelled LCD hemiprotein. The labelled LCD was recruited to ANXA11 FL-GUVs and not ARD-GUVs (Fig. 3C). This recruitment is thus likely underpinned by co-condensation of the exogenous LCD hemiprotein with the LCD of the full-length ANXA11 (as seen in Supplementary Fig. 3A). Additionally, the exogenous LCD decreased the mobility of ANXA11 FL on GUVs (Fig. 3d). This suggests that the addition of LCD-hemiprotein can drive a phase transition of membrane-bound ANXA11 into a condensed, gel-like state. In line with this concept, no change in ARD mobility occurred following the addition of LCD hemiprotein to ARD-GUVs (Fig. 3d).

We next used our PK dye to determine whether this condensed ANXA11 phase resulted in changes in the underlying lipid phase state (Fig. 3e). We have demonstrated above (Fig. 2d, e) that ANXA11 FL and the ARD hemiprotein exert comparable effects on lipid phase state at low protein concentrations. However, the induction of ANXA11 FL condensation (through the addition of exogenous LCD) causes a significant further increase in lipid order (Fig. 3f). This effect is not observed on ARD-GUVs (Fig. 3f).

This ANXA11-mediated effect on the lipid phase state could in principle arise by two mechanisms. Firstly, a change in the phase state of the LCD of the full length ANXA11 might indirectly impact the lipids through altering ARD-based interactions with lipid headgroups. Alternatively, a phase transition of the LCD in close proximity to lipid membranes could prove sufficient by itself to directly influence the lipid phase state. To disambiguate these two possibilities, we constructed an experimental system in which LCD hemiprotein could engage with GUV membranes in the absence of the ARD. To accomplish this, we spiked our GUVs with 10% DGS-NTA(Ni), a nickel-chelating lipid species that binds to histidine. We then purified an LCD hemiprotein containing a C-terminal 6x HIS tag. This C-terminal HIS tag permits the LCD to associate with the lipid surface in the absence of the ARD (Fig. 3g). A HIS-tagged Halo protein was used as a surface binding control which cannot undergo condensation. PK dye readouts from the GUV membranes reveal a modest but significant increase in lipid order following binding of the HIS-LCD protein, but not the HIS-Halo control (Fig. 3h, i). This indicates that phase coupling between ANXA11 and lipid membranes can occur independently from a direct contribution of the ARD. Of note, by lowering the amount of DGS-NTA(Ni) in our GUV membranes to 2.5% such that only a small fraction of membrane lipids can engage with HIS-tagged LCD, we observed segregation of LCD into dilute and dense phases on the same membrane surface (Supplementary Fig. 3F). HIS-tagged Halo does not segregate and instead uniformly coats the GUV membrane. These data further support the capacity of LCD to undergo condensation on the lipid surface, while the Halo control cannot.

Taken together, these findings suggest that the ARD-mediated liquid-to-gel lipid phase transition described in Fig. 2 can be tuned by condensation of the ANXA11 LCD, thus providing evidence of protein-lipid phase coupling within ANXA11-GUV assemblies.

## ANXA11 interactors ALG2 and CALC modulate phase coupling within ANXA11-GUV assemblies

Prior work on other annexins has revealed an assortment of functional interaction partners[40]. We wondered whether similar binding partners might exist for ANXA11, and if so, whether they could modulate ANXA11's phase state and also impact phase coupled changes in underlying lipid membranes. A bioinformatics search revealed two candidate proteins, ALG2 and CALC, which bind to distinct small, structured motifs within the ANXA11 LCD.

Initially we co-incubated FL ANXA11 with either purified ALG2 or CALC (Supplementary Fig. 4A) at molar ratios matching their relative abundance in cells (1[ANXA11]:4[ALG2]:40[CALC])[41]. Under these conditions, neither ALG2 nor CALC form condensates by themselves (Supplementary Fig. 4B). However, when mixed with ANXA11, both ALG2 and CALC co-partition into FL ANXA11 condensates, unlike a Halo control protein (Supplementary Fig. 4C). Both modulators dramatically alter the dispersed-to-condensed phase boundary of ANXA11. However, they do so in opposite directions (Supplementary Fig. 4D). ALG2 significantly increases ANXA11 condensation, while CALC has the opposite effect (Fig. 4a). Our FRAP studies revealed that the addition of ALG2 also pushes ANXA11 condensates into a more immobile gel-like state (Supplementary Fig. 4E). Of note, this transition does not elicit changes to the secondary structure of ANXA11 that indicate protein aggregation (Supplementary Fig. 4F, G). By contrast, CALC increases the mobility of ANXA11 condensates, driving them into a more fluid state (Supplementary Fig. 4E). Once again, the addition of a Halo control protein had no effect on the ANXA11 phase boundary (Supplementary Fig. 4D), condensate formation (Fig. 4a), or ANXA11 mobility (Supplementary Fig. 4E). These results infer that protein crowding is unlikely to be a driving factor in the observed influence of ALG2 or CALC on ANXA11.

Having established ALG2 and CALC as robust antipodal regulators of ANXA11 phase state in solution, we next investigated their influence on ANXA11-coated lipid membranes. By themselves, neither ALG2 nor CALC associate with GUV membranes (Supplementary Fig. 5A). However, both modulators are recruited to the surface of GUVs when FL ANXA11 is present (Supplementary Fig. 5A). Here, ALG2 acts to increase FL ANXA11 recruitment to GUVs, while CALC decreases it (Fig. 4b, c). These effects are likely mediated through the aforementioned influence of ALG2 and CALC on LCD-based ANXA11 condensation. In support of this, neither ALG2 nor CALC altered recruitment of ARD hemiprotein to GUVs (Supplementary Fig. 5B, C). By contrast, the recruitment of HIS-LCD hemiprotein to DGS-NTA(Ni)-containing GUVs (as described in Fig. 3g) was increased by ALG2 and impaired by CALC (Fig. 4d). These differential effects of ALG2 and CALC on FL ANXA11

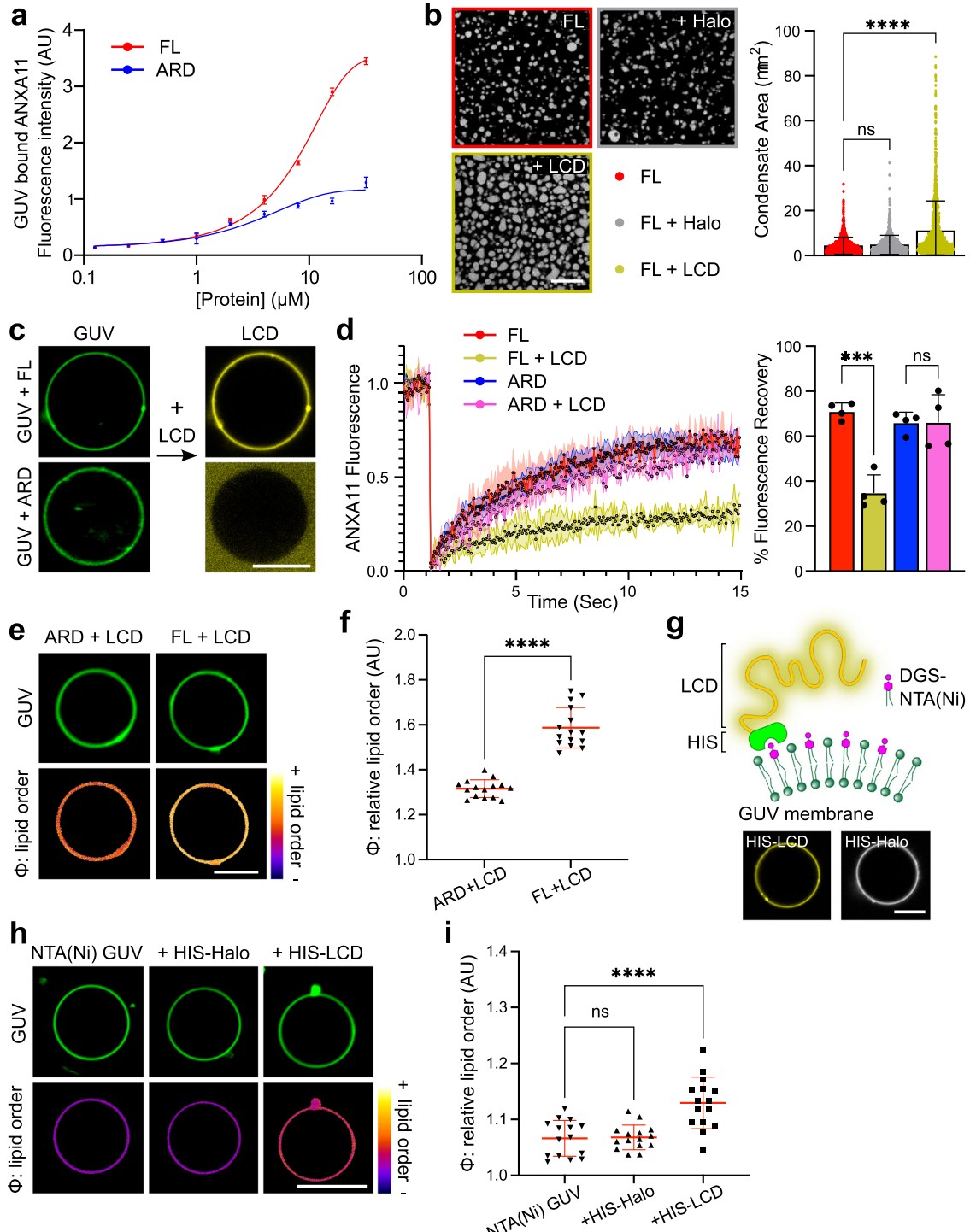

**Fig. 3 | ANXA11 condensation induces coupled changes in the phase state of lipids in GUV membranes. a** Log(dose) response curves fitted for AF647-labelled ANXA11 FL and ARD binding to GUV membranes at 100 μM Ca²⁺. Mean ± SD, *n* = 3. **b** Fluorescence micrographs of 25 μM recombinant AF647-ANXA11 FL alone or with unlabelled 25 μM LCD or 25 μM Halo. Scale bar–5 μm. Corresponding quantification for condensate area is plotted alongside. Mean ± SD, Kruskal-Wallis with Dunn's multiple comparison, ****p < 0.0001, ns–not significant (p > 0.05), *n* = 3 repeats (1171–1439 condensates). **c** Representative images of ATTO488 GUVs coated in either 0.5 μM ANXA11 FL or 0.5 μM ARD at 100 μM Ca²⁺ (left). On the right, the same GUVs coincubated with 25 μM AF647-labelled LCD. Scale bar–5 μm. **d** FRAP recovery curves of 0.5 μM AF647-labelled ANXA11 FL or ARD on the surface of GUVs at 100 μM Ca²⁺, with and without 25 μM LCD. The corresponding quantification of the % fluorescence recovery after 15 s is plotted alongside. Mean ± SD. One-way ANOVA with Tukey's multiple comparison, ***p = 0.001, ns–not significant (p > 0.05), *n* = 4

GUVs. **e** Fluorescence images of ATTO647 GUVs labelled with 5 μM PK dye to extract the relative order (φ) of membrane lipids. GUVs were incubated with 100 μM Ca²⁺ and either 0.5 μM ANXA11 or ARD in the presence of 25 μM LCD. Scale bar–5 μm. **f** Quantification of the relative lipid order (φ) of GUVs as shown in (**e**). Mean ± SD. Unpaired two-tailed with Welch's correction, ****p < 0.0001, *n* = 5 repeats (44–126 GUVs). **g** A schematic illustrating HIS-LCD conjugation to DGS-NTA(Ni) lipids. Below are images showing 25 μM AF647-labelled HIS-LCD or HIS-Halo binding to NTA(Ni) GUVs. **h** Fluorescence images of ATTO647 GUVs labelled with 5 μM PK dye to extract the relative order (φ) of membrane lipids. NTA(Ni) GUVs were incubated with either 25 μM HIS-LCD or HIS-Halo. Scale bar–5 μm. **i** Quantification of the relative lipid order of GUVs as shown in (**h**). Mean ± SD. One-way ANOVA with Dunnett's multiple comparison, ****p < 0.0001, ns–not significant (p > 0.05), *n* = 5 repeats (83–114 GUVs).

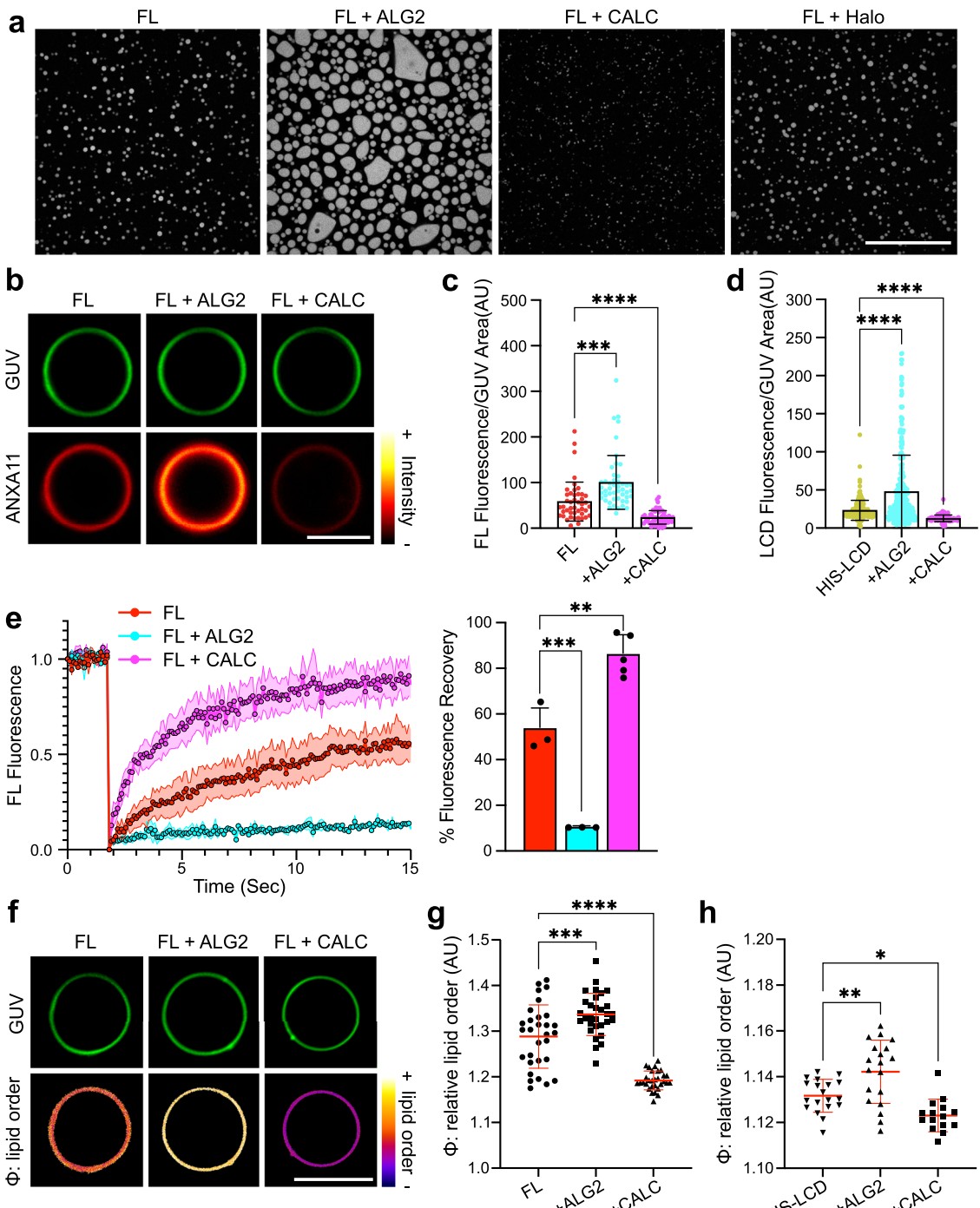

**Fig. 4 | ALG2 and CALC modulate phase coupling within ANXA11-GUV assemblies. a** Fluorescence micrographs of 25 μM recombinant AF647-ANXA11 FL in the presence of unlabelled 0.1 mM ALG2, 1 mM CALC or 1 mM Halo. The concentrations of the modulators were determined from molar ratios matching their relative abundance in cells (1[ANXA11]:4[ALG2]:40[CALC]). Scale bar–20 μm. **b** Representative fluorescence images of ATTO488 GUVs incubated with 0.5 μM AF647-ANXA11 FL at 100 μM Ca$^{2+}$ co-incubated with either 2 μM ALG2 or 20 μM CALC. Scale bar–5 μm. **c** Quantification of the fluorescence intensity of AF647-ANXA11 FL recruited to GUVs as in (**b**). Mean ± SD. Kruskal-Wallis test with Dunn's multiple comparison, ***$p = 0.0002$, ****$p < 0.0001$, $n = 3$ repeats (35–55 GUVs). **d** Quantification of the fluorescence intensity of 25 μM AF647-labelled HIS-LCD recruited to NTA(Ni)-GUVs co-incubated with either 0.1 mM ALG2 or 1 mM CALC. Mean ± SD. Kruskal-Wallis test with Dunn's multiple comparison, ****$p < 0.0001$, $n = 3$ repeats (117–281 GUVs). **e** A FRAP recovery profile of 0.5 μM AF647-ANXA11 FL

on the surface of GUVs in the presence or absence of 2 μM ALG2 or 20 μM CALC at 100 μM Ca$^{2+}$. The corresponding quantification of the % fluorescence recovery after 15 s is plotted alongside. Mean ± SD. One-way ANOVA with Dunnett's multiple comparison ***$p = 0.0004$, **$p = 0.0011$, $n = 3$ GUVs (FL;FL + ALG2) 5 GUVs (FL + CALC). **f** Fluorescence images of ATTO647 GUVs labelled with 5 μM PK dye to extract the relative order (φ) of membrane lipids. GUVs were incubated with 100 μM Ca$^{2+}$ and 0.5 μM ANXA11 FL co-incubated with either 2 μM ALG2 or 20 μM CALC. Scale bar–5 μm. **g** Quantification of the relative lipid order (φ) of GUVs shown in (**f**). Mean ± SD. One-way ANOVA with Dunnett's multiple comparison, ***$p = 0.0006$, ****$p < 0.0001$, $n = 5$ repeats (58–134 GUVs). **h** Quantification of the relative lipid order (φ) of NTA(Ni)-GUVs labelled with 5 μM PK dye. NTA(Ni) GUVs were incubated with 25 μM HIS-LCD and with either 0.1 mM ALG2 or 1 mM CALC. Mean ± SD. One-way ANOVA with Dunnett's multiple comparison, *$p = 0.0318$, **$p = 0.0045$, $n = 4$ repeats (28–94 GUVs).

recruitment are accompanied by parallel changes in ANXA11 mobility on GUVs as measured by FRAP. ALG2 reduces and CALC increases FL ANXA11 mobility on GUV surfaces (Fig. 4e). Neither modulator protein influences the mobility of ARD hemiprotein (Supplementary Fig. 5D). These data suggest that ALG2 promotes ANXA11 condensation on the surface of GUVs, driving ANXA11 into a more gel-like state. CALC has the opposite effects of minimising ANXA11 condensation and driving the protein into a more fluid state.

Using PK dye labelled GUVs, we next looked at whether ALG2- or CALC-based changes in ANXA11 condensation resulted in coupled changes in the underlying lipid phase state. The addition of ALG2 to FL ANXA11-coated GUVs increased lipid order, while CALC decreased it (Fig. 4f, g). Crucially, ALG2 and CALC alone do not impact the lipid phase state, and their effects require the ANXA11 LCD (Supplementary Fig. 5E). Indeed, ALG2 and CALC can modulate the phase state of lipids through HIS-LCD hemiprotein alone when directly conjugated to DGS-NTA(Ni)-containing GUVs (Fig. 4h). This demonstrates that the effects of ALG2 and CALC on ANXA11-based phase coupling do not require the ARD. Control experiments using HIS-Halo protein in place of HIS-LCD reveal no effect on lipid ordering, and no modulation by ALG2 or CALC (Supplementary Fig. 5F). Collectively, this set of experiments identify ALG2 and CALC as putative physiological regulators of ANXA11-lipid phase coupling.

### Effects of ALG2 and CALC on phase coupling in ANXA11-lysosome complexes in living cells

We next evaluated whether phase coupling between ANXA11 and lipid membranes occurs in living cells. To do this, we investigated whether expression of ANXA11 with either ALG2 or CALC altered the lipid phase state of lysosomal membranes in U2OS cells.

Lysosomes are highly dynamic organelles that are typically small and often close to or beneath the resolution limit of standard fluorescence microscopes. Resolving lysosomal membranes and their lipid properties using optical based approaches is therefore challenging. To circumvent these issues, we briefly exposed U2OS cells to a hypotonic buffer to both arrest lysosomal motion and induce lysosomal swelling. Previous work has demonstrated this approach as a powerful method for studying membrane organisation and lipid phase behaviour inside living cells[42–44]. It is important to note that exposure to hypotonic conditions might change the morphology of ANXA11 condensates and likely alters the precise lipid ordering of lysosomal membranes. Consequently, these results should be interpreted comparatively across conditions, and not as absolute measurements.

We initially validated this approach in U2OS cells by showing that lysosomes retain their morphology and lumenal content upon swelling (Supplementary Fig. 6A). Under these conditions, fluorescently labelled ANXA11 also retains its previously reported[16] localisation to lysosomal membranes (Fig. 5a). We then applied the same PK dye method as above to assess lipid ordering in lysosomal membranes of cells transfected with either ANXA11 alone, or ANXA11 + ALG2, or ANXA11 + CALC. Expression levels of the relative proteins were elevated as expected following transfection (Supplementary Fig. 6B). Because the PK dye non-specifically labels all membranes, these experiments were conducted in the presence of a lysosomal membrane marker to permit lysosome-specific PK dye analysis (see live cell microscopy PK dye analysis Methods section, Fig. 5b). The expression of ALG2 or CALC alone, without ANXA11, exert no detectable effect on lipid order in lysosomal membranes (Supplementary Fig. 6C). However, co-expression of ALG2 together with ANXA11 significantly increases the order of lysosomal membranes (Fig. 5c), thereby confirming that the ANXA11-mediated phase coupling effects observed in biochemical reconstitution systems can also occur in cells. Co-expression of CALC with ANXA11 also does not

significantly alter lysosomal lipid ordering. This is likely because under basal conditions lysosomal membranes exist in a relatively disordered state[45]. As such, the influence of CALC is not detectable under these experimental conditions (Fig. 5c).

### ALG2 and CALC modulate the nanomechanical properties of ANXA11-lipid assemblies and their ability to engage RNP granules

ANXA11 plays an important role in long-distance transport of RNP granules in neurons by tethering them to lysosomes[16]. We wondered whether the effects of ALG2 and CALC on ANXA11-lipid phase coupling might: (i) alter the nanomechanical properties of the ANXA11-lysosome ensemble, perhaps to resist disruption by shear stress during rapid axon transport[46], or (ii) impact the RNP granule-tethering capacity of the ensemble.

To analyse the resistance of ANXA11-GUV assemblies to shear stress we designed a microfluidic-based mechanical stiffness assay (Fig. 6a). The deformation of GUVs (strain) was recorded under different flow rates (stress) as they were pushed back and forth into V-shaped channels which tapered in diameter (Fig. 6b). We then calculated the relative elastic modulus from the relationship between the applied pressure and the elongation of the GUV (see elastic modulus Methods section)[47]. Binding of ANXA11 to GUVs resulted in a substantial >2-fold increase in the relative elastic modulus (Fig. 6c), reflecting the ARD-based liquid-to-gel lipid phase transition described in Fig. 2. The addition of ALG2 together with ANXA11 caused an even greater increase (>3-fold) in relative elastic modulus. Adding CALC had the opposite effect, reducing the relative elastic modulus closer to that of naked GUVs. As expected, ALG2 and CALC on their own do not impact the elastic modulus because they do not bind GUVs (Supplementary Fig. 5A). In summary, the effects of ALG2- and CALC on phase coupling (as described in Fig. 4) has a profound impact on the nanomechanical properties of ANXA11-lipid assemblies.

To explore the functional impact of ALG2 and CALC on the capacity of ANXA11-GUV assemblies to engage RNP granules, we established an in vitro RNP granule binding assay (see cell free RNP granule-lipid binding assay Methods section). First, we isolated RNP granules from live U2OS cells using fluorescence activated particle sorting (FAPS)[48]. This process involved: (i) culturing of a stable U2OS line expressing a fluorescent RNP granule marker (G3BP1-mEmerald, Supplementary Fig. 7A); (ii) inducing RNP granule formation prior to cell lysis; and (iii) FAPS-isolating purified RNP granules (Fig. 6d). Mass spectrometry of the FAPS-isolated material confirmed the presence of numerous proteins previously identified as core components of RNP granules within living cells (Supplementary Data 1).

Co-incubation of the FAPS-purified RNP granules with ANXA11-coated GUVs results in accumulation of the RNP granules onto the GUV surface (Fig. 6e). In contrast, recombinantly purified G3BP1 alone is not recruited to ANXA11-GUVs (Supplementary Fig. 7B). Furthermore, RNP granules do not bind to naked GUVs or to ARD-coated GUVs (Supplementary Fig. 7C, D). Thus, RNP granule recruitment to GUVs requires FL ANXA11 holoprotein. Crucially, the addition of ALG2 increases RNP granule binding to ANXA11-GUVs, while CALC decreases it (Fig. 6f). These experiments reveal that ALG2 and CALC modify the ability of ANXA11 to tether RNP granules to lysosomal phospholipids.

These data suggest that ALG2 and CALC-mediated effects on phase coupling impact the nanomechanical properties of the ANXA11-lysosome ensemble, which correlates with the capacity of the ensemble to engage RNP granules. Together, these functional consequences have important implications for how RNP granule trafficking might be regulated in neurons (discussed below).

## Discussion

The experiments described here uncover a rich crosstalk between the biophysical, chemical and nano-mechanical properties of protein and

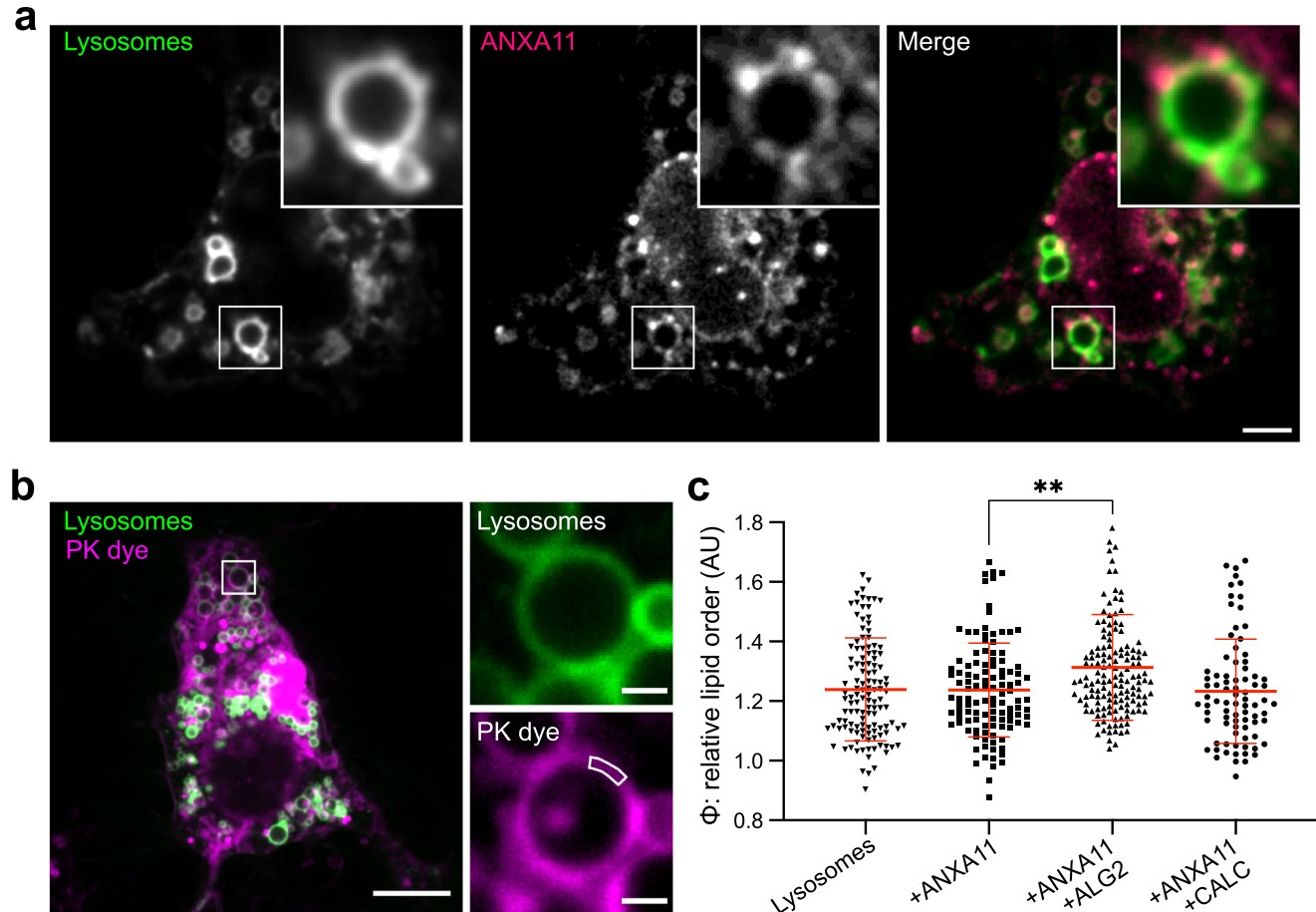

**Fig. 5 | ALG2 promotes ANXA11 condensation to increase the lipid order of lysosomes in cells. a** U2OS cells under hypotonic conditions (95% ddH$_2$O, 5% DMEM, pH 7.0) expressing a fluorescently labelled lysosome marker (TMEM192-Halo-JF647) and ANXA11 (ANXA11-mEm). The cytosolic pool of ANXA11 was photobleached to reveal ANXA11-membrane associations. The white ROI indicates the displayed zoomed regions. Scale bar - 5 μm. **b** A representative image of a hypotonic U2OS cell with fluorescently labelled lysosomes (TMEM192-Halo-JF647)

incubated in 100 nM PK dye to extract the relative order of lysosomal membrane lipids. The white ROI within the zoomed panels indicates an example analysis segment. Scale bars - 10 μm (left panel) and 1 μm(right panel). **c** Quantitation of the relative order (φ) of lysosomal membranes as displayed in (**b**) from U2OS cells alone, or expressing either ANXA11, ANXA11 + ALG2, or ANXA11 + CALC. Mean ± SD. One-way ANOVA with Tukey's multiple comparison, **$p = 0.0021$, $n = 3$ repeats (82–146 lysosomes).

lipid components of juxta-membrane biomolecular condensates and their membrane-bound partners (Fig. 7).

Specifically, we show that ANXA11 undergoes a dispersed to condensed phase transition to form biomolecular condensates, and that this effect requires the N-terminal low complexity domain[16]. In parallel, we show that the binding of the C-terminal annexin repeat domain to lysosomal-like membranes causes a liquid-to-gel phase transition in the underlying lipids. This observation is in agreement with prior work in artificial membranes using ANX5 and ANXA2[24,25,49]. Both ANX5 and ANAXA2 contain four annexin repeat motifs that are homologous to the ANXA11 ARD, and both elicit lipid phase transitions or clustering of specific lipid species in synthetic membranes[24,25,49]. However, ANXA5 and ANXA2 lack intrinsically disordered low complexity domains at their N-termini, and so do not form biomolecular condensates. Consequently, in those systems, it was not possible to explore whether changes in the condensation state of these proteins are coupled to changes in the phase state of the lipid membrane partner.

Here, we took advantage of the presence of the ANXA11 LCD to investigate this possibility. We show that the addition of purified LCD hemiprotein to full length ANXA11 in solution promotes protein condensation, and drives ANXA11 condensates into a less mobile, more gel-like state. On membrane surfaces, ANXA11 condensation tunes the magnitude of the ARD-based liquid-to-gel phase transition in the

subjacent lipids. Crucially, the role of the ANXA11 ARD can be replaced by direct chemical conjugation of the ANXA11 LCD to membrane lipids. In this scenario an increase in lipid membrane order is still observed. This experiment indicates that the phase state of the ANXA11 LCD can be directly coupled with the phase state/ordering of underlying membrane lipids. We use the term phase coupling to denote this phenomenon. A broader role for these mechanisms is supported by recent work describing similar protein-lipid phase coupling at the plasma membrane that plays an important role in T-cell activation[20,22,23]. Unlike models on the plasma membrane, where lipid phase *separation* occurs, we see that ANXA11 binding induces a bulk phase *transition* of lipids into more ordered states across the entire lysosomal membrane. The condensation state of ANXA11 can modulate this bulk lipid transition, again across the entire surface of the membrane. We propose that this subserves a mechanical function, stiffening the entire ensemble to resist sheer stress during axonal trafficking (discussed below).

In this study we also establish that two ANXA11 interacting proteins, ALG2 and CALC, act as potent regulators of phase coupling in ANXA11-lipid assemblies. Specifically, ALG2 promotes condensation of ANXA11, which in turn results in a coupled increase in lipid order. By contrast, CALC impairs ANXA11 condensation which causes a coupled decrease in lipid order. Elucidating the structural mechanisms by which ALG2 and CALC have such differing effects will require further

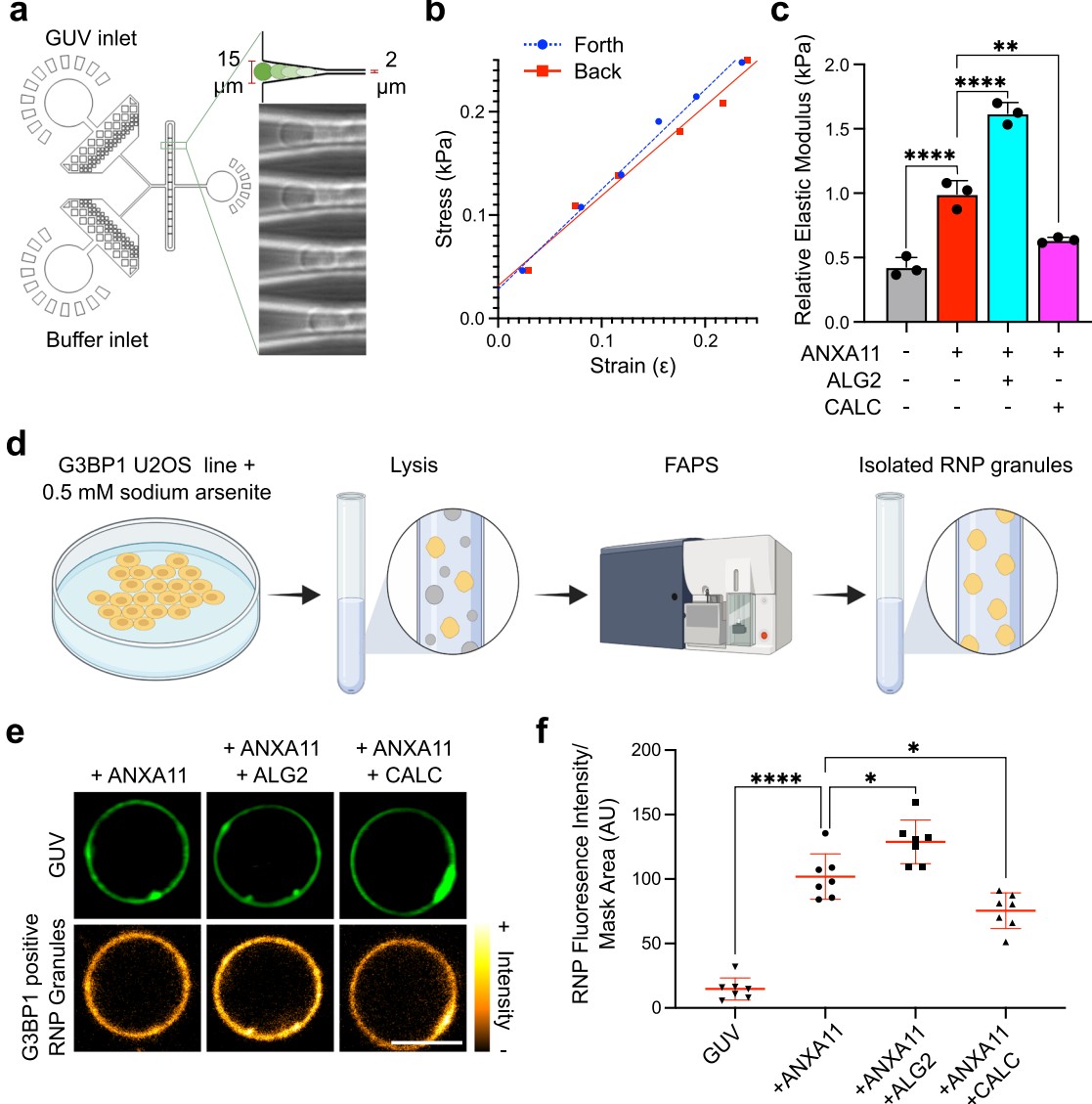

**Fig. 6 | ALG2 and CALC alter the nanomechanical properties of ANXA11-GUV assemblies and their ability to tether RNP granules. a** A Schematic of our microfluidic device used to extract the relative elastic modulus of GUVs. The bright-field images on the right illustrate GUV deformation within the device with a channel size: opening = 15 μm, tapered end = 2 μm. **b** A plot of the GUV deformation (strain) under variable pressure applied across the V-shaped channel (stress). Simple linear regression, $R^2$ (Forth/Back) − 0.991/0.980. ANCOVA-slopes ($p = 0.26$) and intercepts ($p = 0.13$) are not significantly different. **c** Quantification of the relative elastic modulus of GUVs at 100 μM Ca²⁺ with 0.5 μM ANXA11 FL, coin-cubated with either 2 μM ALG2 or 20 μM CALC. Mean ± SD. One-way ANOVA with Tukey's multiple comparison, **$p = 0.0023$, ****$p < 0.0001$, $n = 3$ GUVs. **d** The

experimental pipeline for FAPS-based RNP granule isolation from a stable G3BP1-mEmerald U2OS line. Created in BioRender. Nixon-Abell, J. (2025) https://BioRender.com/j66j626. **e** Representative fluorescence images of ATTO594 GUVs incubated with 100 μM Ca²⁺ and 0.5 μM ANXA11 co-incubated with either 2 μM ALG2 or 20 μM CALC. To each condition, purified RNP granules (labelled with mEmerald-G3BP1) were added to a final concentration of 0.2 mg/ml. Scale bar - 5 μm. **f** Quantification of the fluorescence intensity of RNP granules (mEm-G3BP1) recruited to ANXA11-GUV assemblies as displayed in (**e**). Mean ± SD. One-way ANOVA with Tukey's multiple comparison, *$p = 0.0111$ (A11 vs A11 + ALG2) ;0.0125 (A11 vs A11 + CALC), ****$p < 0.0001$, $n = 7$ repeats (21–78 GUV).

structural studies. However, we speculate that the different locations of their binding motifs within the ANXA11 LCD[27,28] could promote/disrupt the necessary inter- and intra-molecular interactions required for ANXA11 condensation.

Additionally, we have shown that a functional consequence of ALG2/CALC-based alterations in phase coupling is the modulation of the nanomechanical properties of the ANXA11-lipid membrane ensemble. These changes in phase coupling and nano-mechanical properties also correlate with the ability of the ensemble to engage RNP granules. Further experiments will be required to fully understand the biological implications of these changes. However, we speculate that ALG2-based stiffening of the ensemble might enable the complex

to withstand shear stresses during cotransport through the crowded axonal cytoplasm. Indeed, we have recently shown that shear stress itself can increase stiffness of condensates comprised of FUS and RNA (components of RNP granules)[46]. Conversely, CALC, by softening the ensemble, could facilitate on/off-loading of the RNP granule from the ensemble at specific 'pick-up' or 'drop off' locations. Finally, understanding if and how ALS-associated ANXA11 mutations might influence phase coupling might provide additional insights into the pathogenetic mechanisms of ALS-associated ANXA11 mutations[50].

In summary, our study provides new insights into a previously unrecognised mechanism for functional crosstalk between biomolecular condensates and their associated organellar membrane partners.

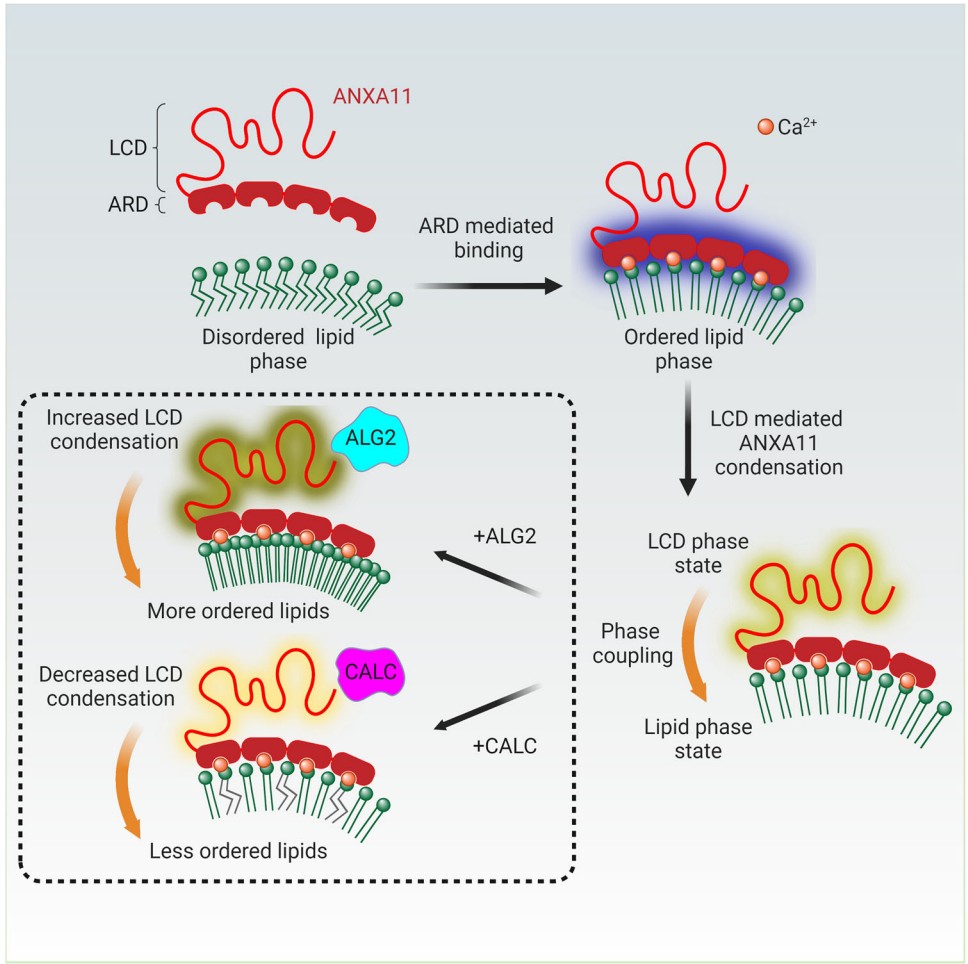

**Fig. 7 | Protein-lipid phase coupling in the ANXA11-lysosome ensemble.** The ARD of ANXA11 mediates binding to lysosomes in a $Ca^{2+}$-dependent manner and causes a phase transition in lysosomal membrane lipids into a more ordered state. Condensation of the ANXA11 LCD can then act to tune the magnitude of this lipid phase transition in a coupled manner. ANXA11 interacting partners ALG2 and CALC either increase (ALG2) or decrease (CALC) ANXA11 LCD-based condensation to regulate phase coupled effects on lysosomal membrane lipids.

It will be of great interest, in future work, to discover whether other biomolecular condensates alter the lipid organisation and functional properties of the diverse group of membrane-bound organelles across eukaryotic cells.

## Methods

### Plasmids and cloning

Constructs encoding ANXA11 FL (aa 1-505), ANXA11 LCD (aa 1-185), ANXA11 ARD (aa 186-505), ALG2, CALC and G3BP1 were cloned by PCR amplification from HeLa cell cDNA into the bacterial expression vector pOPINS (Merck) containing a ULP protease cleavable N-terminal His-SUMO Tag (pOPINS-A11-FL, A11-LCD, ARD, ALG2, CALC, G3BP1: SQ_001-110). For mammalian cell line work, ANXA11 FL and TMEM192-Halo were inserted either side of a T2A self-cleaving peptide in a custom CMV-driven polycistronic vector (ANXA11-T2A-TMEM192-Halo: JNA_215-222). Where co-expression of ALG2 or CALC was required, either gene was inserted into an alternative ORF in the same vector with expression driven using a hPGK promoter (ANXA11-T2A-TMEM192-Halo_hPGK-ALG2, ANXA11-T2A-TMEM192-Halo_hPGK-CALC: JNA_198,216;217,223-229). For experiments requiring ALG2 or CALC expression in the absence of ANXA11, the ANXA11 cassette in ANXA11-T2A-TMEM192-Halo was replaced by either ALG2 or CALC (CMV-ALG2-T2A-TMEM192-Halo, CALC-T2A-TMEM192-Halo: JNA_215,216;384-387). G3BP1-mEmerald was created by subcloning the G3BP1 cassette from the pOPINS-G3BP1 construct into an mEm-N1 backbone derived from the Clontech N1 system (mEm-C1-G3BP1: MSF_001-004). Cloning was performed throughout by PCR amplification using a Q5 polymerase system (NEB) and Gibson Assembly (NEB). All primers are listed in Table 1.

### Expression and purification of proteins

His-SUMO tagged pOPINS constructs (His-SUMO-ANXA11 FL, -LCD, -ARD, -ALG2,-CALC -G3BP1) were transformed into competent *E. coli* BL21(DE3) (NEB) and expressed overnight at 25 °C in TB autoinduction media. Cells were harvested by centrifugation and the cell lysate for protein isolation was produced by subjecting the harvested cells to a high-pressure cell disruption system (Constant systems). Prior to loading on a Ni-Sepharose Advance column (Bioserve), the cell lysate was clarified by high-speed ultracentrifugation at 100,000 g to remove the cell debris. A standard Ni-affinity protein purification protocol was followed and the column eluates containing the protein were pooled, mixed with ULP protease to remove the His-SUMO tags, and dialysed in dilution buffer (25 mM HEPES pH 7.4, 225 mM NaCl) supplemented with 5% (w/v) glycerol. After the cleavage, protein was applied on a second Ni-Sepharose Advance column to remove the His-SUMO tag, followed by size-exclusion chromatography using a Superdex S-75 column (Cytiva) in the same buffer. Purified protein fractions were pooled and concentrated in a spin concentrator (Vivapsin), aliquoted, snap frozen in liquid nitrogen and stored at −80 °C for subsequent use.

**Table 1 | Primers used for cloning as described above**

| Primer Name | Sequence (5'- > 3') | Direction |
|---|---|---|
| SQ_001 | TGAAAGCTTTCTAGACCATTTAAAC | F |
| SQ_002 | GGTACCACCGATCTGTTC | R |
| SQ_003 | GCGAACAGATCGGTGGTACCATGAGCTACCCTGGCTATC | F |
| SQ_004 | AATGGTCTAGAAAGCTTTCAGTCATTGCCACCACAGATC | R |
| SQ_005 | GCGAACAGATCGGTGGTACCATGAGCTACCCTGGCTATCCCCCG | F |
| SQ_006 | AATGGTCTAGAAAGCTTTCATGGGGGCACAGCGGGGGT | R |
| SQ_007 | GCGAACAGATCGGTGGTACCATGACCCAGTTTGGAAGC | F |
| SQ_008 | AATGGTCTAGAAAGCTTTCAGTCATTGCCACCACAGATC | R |
| SQ_009 | GCGAACAGATCGGTGGTACCATGGTGATGGAGAAGCCTAGTCCCC | F |
| SQ_010 | AATGGTCTAGAAAGCTTTCACTGCCGTGGCGCAAGCCC | R |
| JNA_215 | ACTAGTGAGGGCAGGGGAAGTC | F |
| JNA_216 | GGTGGCGACCGGTAGCGC | R |
| JNA_217 | TAGCGCTACCGGTCGCCACCATGAGCTACCCTGGCTATC | F |
| JNA_218 | CTTCCCCTGCCCTCACTAGTGTCATTGCCACCACAGATC | R |
| JNA_219 | ATGGCAGAGATAGGAACCG | F |
| JNA_220 | ACTAGTTGGGCCGGGATTTTC | R |
| JNA_221 | AAAATCCCGGCCCAACTAGTATGGCTGCAGGCGGACGC | F |
| JNA_222 | CCGGTTCCTATCTCTGCCATGGTCGCTACAGGAGGATCCCC | R |
| JNA_198 | TCCGGACTCAGATCTCGAGCTCAAG | F |
| JNA_216 | GGTGGCGACCGGTAGCGC | R |
| JNA_217 | TAGCGCTACCGGTCGCCACCATGAGCTACCCTGGCTATC | F |
| JNA_223 | GCTCGAGATCTGAGTCCGGATCACTTGTAGAGTTCATCCATC | R |
| JNA_224 | CTGTGCCTTCTAGTTGCCAGC | F |
| JNA_225 | GGTGGCGACCGGTGGATC | R |
| JNA_226 | GGGATCCACCGGTCGCCACCATGGCCGCCTACTCTTAC | F |
| JNA_227 | CTGGCAACTAGAAGGCACAGTCATACGATACTGAAGACCATG | R |
| JNA_228 | GGGATCCACCGGTCGCCACCATGGCATGCCCCCTGGAT | F |
| JNA_229 | CTGGCAACTAGAAGGCACAGTTAGCCCTTGAGGGCTTCATTG | R |
| JNA_384 | TAGCGCTACCGGTCGCCACCATGGCATGCCCCCTGGAT | F |
| JNA_385 | CTTCCCCTGCCCTCACTAGTGCCCTTGAGGGCTTCATTG | R |
| JNA_386 | TAGCGCTACCGGTCGCCACCATGGCCGCCTACTCTTAC | F |
| JNA_387 | CTTCCCCTGCCCTCACTAGTTACGATACTGAAGACCATGG | R |
| MSF_001 | CGGCCGCGACTCTAGATCATAATCAGCC | F |
| MSF_002 | AAGTAAAACCTCTACAAATGTGGTA | R |
| MSF_003 | GATCGCTAGATCTATGGTGATGGAGAAGCCT | F |
| MSF_004 | GACTGCAGAATTCTCACTGCCGTGGCGCAAG | R |

Halo protein purification was described previously[51]. Protein purity was routinely assessed by SDS-PAGE and all proteins were purified at or above 95% purity level (Supplementary Fig. 4A).

## Alexa Fluor protein labelling

Alexa Fluor AF647, AF555 and AF488 dyes (Thermo Scientific) were conjugated to proteins using NHS-ester chemistry following standard protocols. Briefly, 2.0 mg/ml of the protein was mixed with 100 µg of the dye and incubated stirring at 4 °C for 4 h. Following the incubation, proteins were run on a Superdex-25 column (Cytiva) to separate the free dye from the conjugated protein-dye complex.

## NMR spectroscopy

Isotopically enriched proteins were expressed and purified as described above. ANXA11 FL was expressed with $^{15}$N enrichment and AXNA11 LCD was expressed with either $^{13}$C/$^{15}$N or $^{13}$C/$^{15}$N/$^{2}$H enrichment for residue specific assignments.

All NMR samples were collected in 25 mM HEPES pH 7.4, 225 mM NaCl with the addition of 2–5% D$_2$O for sample locking. All spectra were collected on a 700 MHz Bruker Avance II+ spectrometer equipped with a 5 mm TCI cryoprobe using Topspin 3.6.0 acquisition software.

$^{1}$H-$^{15}$N BEST-TROSY spectra[52] were collected at 278 K and 308 K for both the 200 µM $^{15}$N-ANXA11 FL and 200 µM $^{15}$N-ANXA11 LCD.

A partial (53% of non-proline amino acids, see Supplementary Fig. 9) assignment of backbone H$_N$, N$_H$ and Cα, Cβ, C' resonances of the 150 µM $^{13}$C/$^{15}$N/$^{2}$H ANXA11 LCD sample was based on the following six 3D BEST-TROSY datasets at 278 K, acquired as pairs to provide own and preceding carbon connectivities. B_trHNCO, b_trHN(CA)CO, b_trHN(CO)CA, b_trHNCA, b_trHN(CO)CACB and b_trHNCACB were obtained as experimental pairs with 1024 complex points in the proton, and 80, 128 complex points in the $^{15}$N and $^{13}$C dimensions, respectively[53,54]. Data sets were collected with 10–50% Non Uniform Sampling (NUS) to reduce acquisition times.

The assignment of a subset of proline residues benefitted from C'-detect experiments collected with a 50 µM $^{15}$N/$^{13}$C ANXA11 LCD sample. In these experiments correlations are made between the N$_H$ and its preceding C' carbon (N$_i$ & C'$_{i-1}$) and their own or preceding Cα$_{i,i-1}$ or Cβ$_{i,i-1}$ shifts. $^{1}$Hα-start, C'-detect sequences c_hcacon_ia, c_hcacon_ia3d, c_hcanco_ia3d and c_hcbcacon_ia3d were used for the N$_i$-C'$_{i-1}$, N$_i$-C'$_{i-1}$Cα$_{i-1}$,

$N_i$-C′$_{i-1}$-C$\alpha_{i-1}$ + $N_i$-C′$_{i-1}$-C$\alpha_i$ and $N_i$-C′$_{i-1}$-C$\beta_{i-1}$ connectivities, respectively[55]. Typical data sets consisted of 1024 complex points in the direct carbon, and 64, 128 complex points in the $^{15}$N and indirect $^{13}$C dimensions, respectively. C′-detect data was processed using Topspin v 3.6.0 (Bruker) and triple resonance NUS data was processed using NMRPipe v10.4[56] including NUS data reconstruction via compressed sensing[57]. Data analysis was completed using NMRFAM-Sparky v3.115, and the assignment tool MARS v1.1.

Secondary structure analysis was completed using TALOS-N where backbone torsion angles are extracted from secondary C$\alpha$, C$\beta$ chemical shifts, i.e. shift differences between experimental data and those seen for the same residue under random coil conditions[58]. Where assignments are missing, TALOS-N can predict secondary structure elements based on sequence alone.

### In vitro protein condensation assays

To visualise protein condensates in vitro, AF647-conjugated purified ANXA11 FL, LCD, ARD, and unlabelled purified ALG2 and CALC were made up in dilution buffer (25 mM HEPES pH 7.4, 225 mM NaCl) to the desired concentrations (listed in figures). Final volumes of 25 μL were transferred to 8-well glass bottom chambered coverslips (Ibidi, 80826) and incubated at 37 °C for 15–30 min before being imaged using a scanning confocal LSM 780 (Zeiss) fitted with a Plan-Apochromat 100× 1.4 NA oil immersion objective (Zeiss). An HeNe 5 mW 633 nm laser line was used for excitation (1 mW max at focal plane), with fluorescence collected on a GaAsP detector in the 620 nm–750 nm λ range. Samples were imaged at 37 °C in a humidified imaging chamber with a Definite Focus module (Zeiss) employed for thermal drift correction and ZEN Black v2.3 (Zeiss) software used for acquisition.

### Generation of phase diagrams

Recombinant ANXA11 FL was mixed on ice in low-protein bind tubes (Eppendorf) with either LCD, ALG2, CALC, or HALO at various molar ratios (listed in figures). 20 μL of each sample was deposited into 8-well glass bottom chambered coverslips (Ibidi, 80826). Before imaging, samples were incubated at 25 °C for 15 min. Sample imaging was performed at 25 °C using Leica Stellaris 5 confocal microscope equipped with a Plan-Apochromat 63× 1.4 NA oil immersion objective (Leica). A Supercontinuum white light laser (440–790 nm) was used for excitation at 653 nm, with fluorescence collected on a Power HyD detector in the 660 nm–750 nm λ range.

Confocal images were collected on LAS X (Leica) software and analysed using FIJI (NIH). The condensate detection threshold for the sample to be considered in the condensed state was defined by (i) the average area of condensates (>0.4 μm$^2$), and (ii) the average number of condensates over a 90 μm$^2$ region (>20 condensates). These parameters were established using the inbuilt particle analysis tool in FIJI.

The data was plotted using Python. The phase boundary was determined by fitting the data using a support-vector machine (SVM) algorithm with a linear or 2nd degree polynomial kernel.

### ANXA11 temperature cycling

For temperature-cycling experiments (Supplementary Fig. 1C, D), AF647 labelled ANXA11 FL was imaged at 50 μM on a custom spinning disk microscope permitting precise and rapid control of sample temperature while allowing rapid imaging. The microscope comprised a Nikon Ti stand equipped with perfect focus, a fast piezo z-stage (ASI), a Plan Apochromat lambda 100X NA 1.45 objective, a Yokogawa CSU-X1 spinning disk head and a Photometrics 95B back-illuminated sCMOS camera. The camera operated in global shutter mode and was synchronised with the spinning disk rotation. Excitation was performed using 637 nm (140 mW OBIS LX) laser fibered within a Cairn laser launch, and a single band emission filter was used (Chroma 655LP). To enable fast acquisition, the entire setup was synchronised at the hardware level by a Zynq-7020 Field Programmable Gate Array

(FPGA) stand-alone card (National Instrument sbRIO 9637) running custom code. In particular, fast z-stacks were obtained by synchronising the motion of the piezo z-stage during the readout time of the cameras. Instrument was controlled by Metamorph software.

For control of temperature, a Cherry Temp stage was used (Cherry Biotech), which allows rapid temperature changes (~5 s) between two set temperatures[59]. The sample was mounted within a chamber formed between a glass coverslip and the Cherry Temp microfluidics chip with a 0.5 mm PDMS insert in between. The sample was cycled through 6 cycles of increasing temperature (5 °C increments) for one minute at a time, interspersed with one-minute periods at 5 °C (to dissolve the condensates). Z-stacks were acquired (ΔZ = 200 nm), and maximum intensity z-projections were computed.

The extent of condensate formation ("granulosity index"), was then evaluated as follows: Raw images were processed for homogenous background subtraction, then a Fast Fourier Transform (FFT) was computed and a high-pass filter was applied via a circular mask prior to an inverse FFT. This mask was kept constant for all images (all source data were cropped to have the same size). We then computed the granulosity index as the ratio between the standard deviation and the mean of the signal in the high pass filtered image.

### GUV and SUV preparation

All lipids were purchased from Avanti Polar Lipids (Alabama, USA). GUVs and SUVs were prepared as described previously[16]. Preliminary experiments were conducted in GUVs composed of 35% POPC, 10% POPS, 15% POPE, 5% SAPI, 5% PI(3)P, 10% cholesterol and 20% sphingomyelin to closely match the composition of lysosomes. However, we found that the overall stability, quality and reproducibility of these GUVs were inadequate. Because the principal binding target of the ANXA11 ARD are phospholipids such as the lysosomally-enriched PI(3)P, we therefore focused our subsequent experiments on GUVs composed of 50% POPC, 10% POPS, 20% POPE, 10% SAPI, 5% PI(3)P, 5% cholesterol.

Briefly, giant unilamellar vesicles (GUVs) were prepared (w/v) from 50% 1-palmitoyl-2-oleoyl-3-phosphocholine (POPC, Avanti 850457), 20% 1-palmitoyl-2-oleoyl-sn-glycero-3-phosphoethanolamine (POPE, Avanti 850757), 10% 1-stearoyl-2-arachidonoyl-sn-glycero-3-phosphoinositol (SAPI, Avanti 850144), 10% 1-palmitoyl-2-oleoyl-sn-glycero-3- phospho-L-serine (POPS, Avanti 840034), 5% 1,2-dioleoyl-sn-glycero-3-phospho-(1′-myo-inositol-3′-phosphate) (PI(3)P, Avanti 850150), 0.05% ATTO488, ATTO594 or ATTO647 labelled 1,2-dioleoyl-sn-glycero-3-phosphoethanolamine (DOPE, Sigma), and 4.95% cholesterol (Avanti 700000) dissolved in chloroform. In certain experiments, 10% 1,2-dioleoyl-sn-glycero-3-[(N-(5-amino-1-carboxypentyl)iminodiacetic acid)succinyl] (nickel salt) (DGS-NTA(Ni), Avanti 790404) was added at the expense of 10% POPC. GUVs were formed via electroformation using a Nanoion Vesicle Prep Pro system (Nanion Technologies, Germany) according to the manufacturer's instructions. Briefly, lipids were dispensed using a Hamilton syringe (Sigma) onto a conductive slide under nitrogen vapour. The chloroform was evaporated from the slide overnight, again under nitrogen vapour, in a dark sealed chamber. The following day, a greased O ring was placed over the lipid film, with lipids then rehydrated in 250 μl of liposome buffer (50 mM HEPES pH 7.4, 450 mM Sucrose). The slide was loaded onto the Vesicle Prep Pro with the following programme details run: Frequency - 10, Amplitude - 1.4, Temperature - 60 °C, Rise - 3 min. Formed GUVs were then removed from the slide and used for future applications.

Small unilamellar vesicles (SUVs) for microfluidic assays were prepared using the same lipid composition as GUVs above. A dry and thin lipid film was prepared by gently evaporating the chloroform using a nitrogen stream. The film was then placed under vacuum overnight to remove all traces of chloroform. Liposome buffer (50 mM HEPES pH 7.4, 450 mM Sucrose) was added to hydrate the lipid film to the concentration of 800 μg/mL and the solution was stirred at room

temperature for at least 2 h. The lipid solution was then frozen and thawed 8 times using liquid nitrogen and a water bath set to 37 °C. The solution was then sonicated on ice with a probe sonicator (3 × 5 min, 50% cycles, 20% maximum power) and centrifuged for 30 min at 6000 $g$ to remove probe residues. The size of SUVs was checked with Dynamic Light Scattering (Zetasizer, Malvern Panalytical, UK). The radius of lipid vesicles was measured to be 18.65 +/- 2.5 nm (9 batches of SUVs produced independently). SUV solution was stored at 4 °C and diluted to 400 μg/mL for the diffusional sizing assay.

### Microfluidic diffusional sizing assays

Microfluidic devices were fabricated using standard soft-photolitography techniques by using a silicone rubber compound (Momentive RTV615, Techsil) mixed with black carbon powder (PL-CB13, PlasmaChem GmbH) to minimise the background signal in fluorescence images. The channel height used was between 25 and 50 μm, with the specific height of each device measured using a Dektak profilometer (Bruker). Upon binding to a glass slide, the devices were filled with dilution buffer (25 mM HEPES pH 7.4, 225 mM NaCl) containing 0.01% v/v Tween 20 (Sigma) at least 1 h prior to the experiment, to prevent protein adhesion to the channel surfaces.

Microfluidic diffusional sizing experiments were performed as described previously[60,61] using the chip design as displayed in Supplementary Fig. 1F. Prior to the sizing experiment, all samples were incubated for 1 h at 37 °C. Proteins were used at 0.5 μM (ANXA11 FL, LCD, ARD), and SUVs at 400 μg/mL in dilution buffer (25 mM HEPES pH 7.4, 225 mM NaCl). Calcium chloride (Sigma; henceforth $Ca^{2+}$) was used at 50, 100 and 500 μM. The sample and buffer were loaded at the inlets, with liquid withdrawn from the outlet using a glass syringe (Hamilton) mounted to a neMESYS syringe pump (Cetoni GmbH). The sample was co-flown with the dilution buffer (also containing appropriate $Ca^{2+}$ concentrations) along the channel at 3–4 different flow rates ranging from 20 to 150 μL/h. For each flow rate, once stabilisation of flow was achieved, fluorescent images were taken of 4 fluidic channels of differing path length corresponding to diffusion times of SUV-protein complexes. The images were obtained with either an inverted Axio Observer A1 microscope (Zeiss) equipped with ET-EGFP (49002, Chroma), ET-CY3/R (49004, Chroma), excitation 628/40 nm, emission 692/40 nm, dichroic mirror 660 nm (Cy5-4040C-000, Semrock) a 5× 0.12 NA A-Plan objective (Zeiss) and an Evolve 512 CCD camera (Photometrics).

The images were analysed with Python (scripts available at [https://zenodo.org/record/3881940#.Y-TgAxPP0bZ]). First, the fluorescence profiles were extracted from the images to determine the spatial distribution of sample molecules as a function of time. Diffusion profiles corresponding to the diffusion coefficients for $R_H$ = 0.1–50 nm, and the best fit to the observed sample distribution at the four time points was used to determine the average $R_H$ for each measurement[62,63].

### Imaging and quantification of protein recruitment to GUVs

To determine the relative recruitment levels of ANXA11 FL, LCD and ARD to GUVs at different $Ca^{2+}$ concentrations, purified AF647-labelled protein was added at a 0.5 μM final concentration to ATTO488-GUVs made up in dilution buffer (25 mM HEPES pH 7.4, 225 mM NaCl). After addition of the desired concentration of $Ca^{2+}$ (listed in figures), 200 μL of the protein-GUV mixture was plated into 8-well glass bottom chambered coverslips (Ibidi, 80826) and incubated at 37 °C for 15–30 min prior to imaging.

To examine the direct recruitment of AF488-conjugated ALG2 and CALC to GUVs, 2 μM ALG2 or 20 μM CALC were incubated in the presence of 100 μM $Ca^{2+}$ with ATTO647 GUVs alone, or co-incubated with 0.5 μM ANXA11 FL (unlabelled) (Supplementary Fig. 5A). To determine the influence of ALG2 and CALC on ANXA11 FL recruitment to GUVs the same experiment was performed using ATTO488-GUVs, AF647-labelled ANXA11 FL and unlabelled ALG2 and CALC.

To quantify the relative abundance of protein recruited to the GUV surface, confocal microscopy was carried out using an LSM 780 (Zeiss). Excitation was performed sequentially using an Argon multi-line 35 mW 488 nm laser (3 mW max at focal plane) and HeNe 5 mW 633 nm laser (1 mW max at focal plane), with imaging conditions experimentally selected to minimise crosstalk. The resulting fluorescence was collected using a 63× Plan-Apochromat 1.4 NA oil immersion objective (Zeiss) and detected on a 34-channel spectral array detector in the 460–600 nm and 620–750 nm λ range. Samples were imaged at 37 °C in a humidified imaging chamber with a Definite Focus module (Zeiss) employed for thermal drift correction and ZEN Black v2.3 (Zeiss) software used for acquisition. To measure protein recruitment to GUVs, data was analysed in FIJI (NIH) using a custom macro. The GUV channel was initially gaussian blurred (sigma radius 1.5) to reduce detector noise, thresholded by intensity, and masked. The integrated fluorescence intensity of the protein channel within the GUV-mask was calculated and normalised to the mask area.

### Collective binding assays

To explore collective binding of ANXA11 to GUVs, purified AF647-labelled ANXA11 FL or ARD were added at varying concentrations (listed in figures) to ATTO488-GUVs made up in dilution buffer (25 mM HEPES pH 7.4, 225 mM NaCl) alongside 150 μM $Ca^{2+}$. Samples were incubated at 37 °C for 30 min before being spun at 9100 g for 10 min. Supernatants were discarded and GUV pellets resuspended in 100 μl of dilution buffer before being transferred to a black flat bottom 96 well plate (Greiner). Fluorescence reads were taken on a Tecan SPARK plate reader (633/20 nm excitation, 660/20 nm emission, 40 μs integration) to measure the relative quantities of GUV-bound ANXA11 FL or ARD.

### Fluorescence recovery after photobleaching (FRAP)

To observe the mobility of ANXA11 FL and ARD on the surface of GUVs, in the presence or absence of 25 μM LCD, or 2 μM ALG2, or 20 μM CALC, FRAP of 0.5 μM AF647-conjugated ANXA11 FL and ARD was performed. Labelled FL or ARD were added to 20 μM ATTO488 GUVs in dilution buffer (25 mM HEPES pH 7.4, 225 mM NaCl) in 100 μM $Ca^{2+}$. 200 μL of the protein-GUV mixture was then plated into 8-well glass bottom chambered coverslips (Ibidi, 80826) and incubated at 37 °C for 15–30 min prior to imaging.

To test the effect of ANXA11 on lipid mobility, 20 μM GUVs in which the PI(3)P was replaced with fluorescent BODIPY-PI(3)P (Echelon) were incubated with or without 0.5 μM AF647-labelled ANXA11 FL for 15–30 min at 37 °C in 100 μM $Ca^{2+}$ as described above.

FRAP experiments were performed on a scanning confocal LSM 780 (Zeiss) fitted with a Plan-Apochromat 100× 1.4 NA oil immersion objective (Zeiss) at 37 °C in a humidified imaging chamber. An Argon multi-line 35 mW 488 nm laser (3 mW max at focal plane) and HeNe 5 mW 633 nm laser (1 mW max at focal plane) were used for excitation and bleaching of GUVs and AF647-labelled protein respectively. Fluorescence was detected on a 34-channel spectral array detector in the 460–600 nm (GUV) and 620–750 nm (protein) λ range. Samples were stabilised for thermal drift using a Definite Focus module (Zeiss) and ZEN Black v2.3 (Zeiss) software was used for acquisition. Images were collected at 11 Hz. FRAP data were plotted using the FRAP Profiler plugin on FIJI (NIH).

For FRAP studies on condensates, AF647-ANXA11 FL was co-incubated with either recombinant LCD, or ALG2, or CALC, or Halo (see figures for concentrations). Imaging conditions were identical to protein FRAP on the surface of GUVs but with the FRAP region drawn internally within a condensate, and imaging performed at 1 Hz.

### In vitro membrane lipid order assays

To assess lipid membrane order using a solvatochromic pyrene probe (PK dye), GUV and protein coincubations were performed as described above (Imaging and Quantification of Protein Recruitment to GUVs),

with the exception that DOPE within the GUVs was labelled with ATTO647 and the protein components were unlabelled. Specific protein, GUV and PK dye concentrations are detailed in the figure legends.

Imaging was performed on a scanning confocal LSM 780 (Zeiss) fitted with a Plan-Apochromat 100× 1.4 NA oil immersion objective (Zeiss) at 37 °C in a humidified imaging chamber. A 30 mW continuous wave 405 nm diode pumped solid state (DPSS) laser (<3 mW max at focal plane) and HeNe 5 mW 633 nm laser (1 mW max at focal plane) were used for excitation of the PK dye and GUVs respectively. Fluorescence was detected on a 34-channel spectral array detector across three spectral windows: 449–550 nm (BFP/GFP), 550–650 nm (RFP), and 650–750 nm (Cy5). Samples were stabilised for thermal drift using a Definite Focus module (Zeiss) and ZEN Black v2.3 (Zeiss) software was used for acquisition.

The Cy5 emission window was used to identify ATTO647 GUVs, with the ratio of the PK dye fluorescence from the BFP/GFP and RFP signals used to calculate the relative lipid order ($\varphi$) in FIJI (NIH), as reported in Valanciunaite et al., 2020. To minimise the contribution of low signal background pixels, a mask was generated from the GUV channel, with regions outside of the mask in the BFP/GFP/RFP channels set to a grey value of 0.

## Elastic modulus
The relative elastic modulus of GUVs was measured in a two-layered microfluidic platform with the channel design shown in Fig. 6a. A rectangular-shaped main channel with height 25 μm was connected by 14 V-shaped tapered small bridges with height 4 μm. The microfluidic system was fabricated on the silicon wafer by exposing once with a film mask and another time with a chrome mask to obtain channels with different heights. Once the GUVs were loaded and trapped in the small V-shaped bridges, different pressure drops (100–1000 Pa) were applied across the bridges by tuning the flow rate of the buffer as controlled by a neMESYS syringe pump (Cetoni). The relationship between the pressure drop across the V-shaped channels and the elongation of the vesicle reveals the elastic property of the GUV. Relative elastic moduli were calculated as described previously[47]. Measurements for the deformation of the GUV were obtained using an AxioObserver inverted microscope (Zeiss) coupled to an Evolve 512 CCD camera (Photometrics). The images were analysed on FIJI (NIH).

## Infrared Nanospectroscopy (AFM-IR)
To perform the AFM-IR spectroscopy, we deposited ANXA11-GUV assemblies (prepared as described above) onto hydrophobic ZnSe surfaces. Our initial AFM survey revealed that these assemblies are heterogeneous, forming micron-sized fragments upon deposition. Some contained only GUV lipid fragments, with no protein present. Other assemblies contained GUV lipid fragments bound to ANXA11. We thus exploited this heterogeneity to compare, in the same sample, the morphological, and chemical properties of GUV lipids without protein versus GUV lipids coated with protein.

We used a nanoIR2 platform (Anasys), which combines high resolution and low-noise AFM with a tunable quantum cascade laser (QCL) with top illumination configuration[64,65]. The sample morphology was scanned by the nanoIR system, with a line rate within 0.1–0.4 Hz and in contact mode. A silicon gold coated PR-EX-nIR2 (Anasys) cantilever with a nominal radius of 30 nm and an elastic constant of about 0.2 N m$^{-1}$ was used.

Both infrared (IR) spectra and images were acquired by using phase loop (PLL) tracking of contact resonance, the phase was zeroed to the desired off-resonant frequency on the left of the IR amplitude maximum and tracked with an integral gain I = 0.1 and proportional gain P = 5[66,67]. All images were acquired with a resolution between 1000 × 500 and 1000 × 100 pixels per line.

The AFM morphology maps were treated and analysed using SPIP software. The height images were first order flattened, while IR maps were flattened by a zero-order algorithm (offset). Nanoscale-localised spectra were collected by placing the AFM tip on the GUV fragments. A laser wavelength of 2 cm$^{-1}$ and a spectral speed of 100 cm$^{-1}$/s within the range 1400–1794 cm$^1$ were used for sampling. Within a single GUV fragment, the spectra were acquired at multiple nanoscale localised positions. Each spectrum collected was averaged 5 times. Successively, the spectra were smoothed by an adjacent averaging filter (3pts) and a Savitzky-Golay filter (second order, 13 points) and normalised. Spectra second derivatives were calculated, smoothed by a Savitzky-Golay filter (second order, 13 points).

GUV fragments containing lipids only possessed typical C=O peaks for phospholipids and cholesterol esters at 1730–32 cm$^{-1}$ (with a spectral width of ~20 cm$^{-1}$ by second derivative analysis) and CH$_2$ group peaks at 1450–1475 cm$^{-1}$ [37,38].

The secondary structural organisation of protein components was evaluated by integrating the area of the different secondary structural contributions in the amide band I, as previously shown[35,68]. The error here was calculated over the average of at least 5 independent spectra. Spectra were analysed using the microscope's built-in Analysis Studio (Bruker) and OriginPRO (OriginLab). All measurements were performed at room temperature, with laser powers between 0.1 and 0.5 mW and under controlled Nitrogen atmosphere with residual real humidity below 5%.

## Fourier Transform Infrared Spectroscopy (FTIR)
Attenuated total reflection Fourier transform infrared spectroscopy (FTIR) was performed using a Bruker Vertex 70 spectrometer equipped with a diamond ATR element. The resolution was 4 cm$^{-1}$ and all spectra were processed using OriginPRO software (OriginLab). The spectra were averaged (3–19 spectra with 256 co-averages), smoothed applying a Savitzky-Golay filter (2nd order, 9 points) and then the second derivative was calculated applying a Savitzky-Golay filter (2nd order, 11 points).

## Cell culture and treatments
U2OS cells (ATCC) were cultured in Dulbecco's Modified Eagle Medium (DMEM, Gibco) with 10% fetal bovine serum (FBS, Gibco) and seeded in triplicate into Matrigel-coated 8-well glass bottom chambered coverslips (Ibidi, 80826). At ~70% cell confluency, cells were transiently transfected using Lipofectamine 3000 (ThermoFisher) and 1 μg of the indicated plasmid according to the manufacturer's instructions. Cells were subsequently imaged ~24 h after transient transfection.

For the PK dye experiments, U2OS cells were transiently transfected with the plasmids indicated as described above. 24 h after transfection, cells were washed two times with PBS. To label Halo-TMEM192 on lysosomal membranes, the JF646 Halo ligand (a kind gift from the Lavis Lab) was diluted to 1:2000 in optiMEM (Gibco) and added to the cells for 1 min at 37 °C and 5% CO$_2$. Cells were rinsed three times with PBS and 100 nM PK dye diluted in DMEM with 10% FBS was applied to the cells for 10 min. Cells were then rinsed three times with PBS and hypotonic buffer (95% ddH$_2$O, 5% DMEM, pH 7.0) was applied to cells. Cells were then incubated for 10 min at 37 °C and 5% CO$_2$ to allow for lysosomal swelling. Cells were subsequently imaged for up to one hour post hypotonic treatment.

For fluorescent labelling of the lysosomal lumen, cells were loaded with fluorescent dextran utilising a three hour pulse of 0.5 mM 10 kDa AF594-dextran (ThermoFisher). After 5x washes with PBS, a chase was performed for three hours in full DMEM with 10% FBS. Cells were then treated with hypotonic media as above and imaged.

## Live cell microscopy PK dye analysis
Live cell microscopy was performed with a customised Nikon TiE inverted microscope utilising a Yokogawa spinning disk scan head (CSU-X1, Yokogawa) and a Prime BSI sCMOS camera (Photometrics).

Cell fluorescence was collected with 525/36 m (GFP), 605/70 m (RFP) and 700/75 m (Cy5) filter sets (Chroma) through a 100× Plan-Apochromat 1.4 NA oil immersion objective (Nikon). Cells were maintained at 37 °C and 5% $CO_2$ with a Tokai Hit stage-top incubator. For experiments interrogating lysosomal lipid order, PK dye signal was imaged with 405 nm laser excitation. Data were acquired in two sequential scans through the GFP and RFP filter sets, with a 200 ms exposure time.

Images from experimental replicates were pooled and regions of intracellular lysosome membranes with TMEM192 signal were selected from cells. The ratio of the PK dye fluorescence from the GFP and RFP signals for the regions was calculated after subtraction of background signal. Background signal was calculated for each image by selecting a region of the image plane with no membrane fluorescence in the PK dye channels and averaging the counts. Region selections were performed with FIJI (NIH) and ratios were calculated in Microsoft Excel. It is important to note that exposure to hypotonic conditions likely alters the lipid ordering of membranes and so results should be interpreted comparatively across conditions, and not as absolute measurements.

### Cell free RNP granule-lipid binding assay

RNP granules were isolated from U2OS cells stably expressing G3BP1-mEmerald using fluorescence-activated particle sorting (FAPS). FAPS-based RNP granule isolation has been described in detail previously[48]. Briefly, stress granules, a class of RNP granule, were induced in cells with 0.5 mM sodium arsenite (sigma) for 30 min prior to lysis. Cell pellets were then suspended in lysis buffer (50 mM Tris-HCl, 1 mM EDTA, 150 mM NaCl, 1% NP-40, pH 7.4) with freshly added cOmplete EDTA-free protease inhibitor cocktail (Roche, Merck) and 80 U/mL RNaseOut ribonuclease inhibitor (Promega). Nuclei and cell debris were cleared from the lysate by centrifugation at $200\,g$ for 5 min. Supernatants were further spun at $10,000\,g$ for 7 min, and then pellets were resuspended into lysis buffer in the presence of 80 Units of RNaseOut (Promega). Particles in the pellets were sorted on a High-Speed Influx Cell Sorter (BD Biosciences). Stress granules were isolated based on their size and fluorescence, as detected with 488 nm excitation. To analyse protein composition of the isolated stress granules, the sample was firstly digested by trypsin, and then further processed and analysed using LC-MS/MS by the CIMR proteomics facility (Cambridge Institute for Medical Research, University of Cambridge, UK). See Mass Spectrometry below for details. To verify that the isolated particles were indeed stress granules, we compared our proteomic data with an RNP granule proteomic database (Supplementary Data 1)[69].

To measure the association of purified RNP granules with ANXA11-GUV assemblies in the presence and absence of ALG2 and CALC, proteins and ATTO594 GUVs were prepared as above, with specific concentrations listed within the figure legends. 0.2 mg/ml of FAPS-isolated RNP granules were then added to each sample. For quantification, the total fluorescence intensity of mEmerald-G3BP1 labelled RNP granules was integrated across and normalised to GUV mask area as described above (Imaging and Quantification of Protein Recruitment to GUVs).

### Mass spectrometry

Samples were reduced, alkylated and digested using the S-Trap protocol (Protifi, Fairport, NY) with proteins digested o/n using trypsin in 50 mM HEPES pH8. Digested peptides were eluted with sequential washes with 50 mM HEPES and 0.2% formic acid/acetonitrile and pooled in 0.5 ml tubes (Protein LoBind, Eppendorf). Tryptic peptides were dried almost to completion and re-suspended in 15 μl solvent (3% MeCN, 0.1% TFA) with 7 μl analysed by LC-MSMS using a Thermo Q Exactive mass spectrometer (Thermo Fischer Scientific) equipped with an EASYspray source and coupled to an RSLC3000 nano UPLC

(Thermo Fischer Scientific). Peptides were fractionated using a 50 cm C18 PepMap EASYspray column maintained at 40 °C with a solvent flow rate of 300 nl/min. A gradient was formed using solvent A (0.1% formic acid) and solvent B (80% acetonitrile, 0.1% formic acid) rising from 3% to 40% solvent B by 90 min followed by a 4 min wash at 95% solvent B. MS spectra were acquired at 70,000 resolution between m/z 400 and 1500 with MSMS spectra acquired at 17,500 fwhm following HCD activation. Data was processed in PEAKS X Pro (Bioinformatics Solutions Inc, Waterloo, Canada). Data was searched against a Uniprot Homo Sapiens database and a database of common contaminants using the PEAKS PTM search engine with 312 built-in modifications and the Spider search engine. PTM searches were performed on spectra with a de novo score greater than 15%.

### Overexpression construct validation

Overexpression constructs were designed to increase the levels of ANXA11, ALG2 and CALC in living cells (see Plasmids and Cloning section). To validate these plasmids, they were transfected in U2OS cells as described above (Cell Culture and Treatments). After ~24 h, cells were scraped and incubated at 4 °C in ice cold lysis buffer (25 mM TRIS pH 7.0, 150 mM NaCl, 1% NP40, 1 mM EDTA, 5% glycerol) containing 1x protease inhibitor (Roche). After 30 min, lysates were spun at $10,000\,g$ for 10 min and supernatants diluted into SDS-loading buffer. Samples were run on an SDS-PAGE 4–12% NuPAGE gel (ThermoFisher), transferred to a PVDF membrane (Millipore), blocked at RT for 1 h in 5% BSA (Sigma) in 1x TBS-T, and probed overnight with antibodies against ANXA11 (10479-2-AP Proteintech), ALG2 (ab133326 Abcam), CALC (10245-1-AP Proteintech) and GAPDH (5174S Cell Signalling). Blots were thoroughly washed in TBS-T, labelled with fluorescent IRDye 680 (LiCor) secondary antibodies, washed again in TBS-T and imaged on an Odyssey CLx (LiCor).

### Image processing and display

Display images of GUVs throughout the manuscript are shown with an LUT that is baseline corrected to reduce the contribution of scattered light in the scope. A gaussian blur (sigma radius 1.5) has been applied after LUT application to remove detector noise and additionally supress background scattered light. These steps yield images from which reliable GUV masks can be generated for subsequent analysis. Display PK dye ratios have been subjected to additional processing steps (detailed in In Vitro Membrane Lipid Order Assays). A gaussian blur (sigma radius 1.5) and LUT baseline correction was also applied to display images of cells in Fig. 5 for the same reasons as above. In some display images linear compression of the LUT results in saturation of certain regions within a field of view. This is necessary to highlight regions of lower signal intensity to the reader.

### Statistics

Where possible, a series of normality tests (Anderson-Darling, D'Agostino & Pearson, Shapiro-Wilk, Kolmogorov-Smirnov) were performed to determine if parametric or non-parametric approaches were appropriate. Certain datasets were determined to be non-normally distributed. This is not surprising given that the fluorescence detection range of photomultiplier tubes (PMTs) on confocal microscopes is limited. As such, a dynamic range was selected to prevent pixel saturation at the expense of detecting low fluorescence signal events, thereby skewing the detection window (sampling) towards the upper tail of the distribution. In these cases, a nonparametric Kruskal-Wallis test was performed with Dunn's multiple comparison to compare datasets. Further details are summarised in Supplementary Table 1, with P-values provided within the figure legends.

### Reporting summary

Further information on research design is available in the Nature Portfolio Reporting Summary linked to this article.

## Data availability

The authors declare that data supporting the findings of this study are available within the paper and its Supplementary Information files. Raw files are available on Figshare [https://doi.org/10.6084/m9.figshare.25746051.v1]. Mass spec data is available at [https://www.ebi.ac.uk/pride/archive/projects/PXD060971]. The partial NMR backbone chemical shift assignment for Annexin A11 LCD is available on the BMRB under accession code 52944 and https://doi.org/10.13018/BMR52944. Source data are provided with this paper.

## Code availability

Python scripts for analysing diffusional sizing data are available at [https://zenodo.org/record/3881940#.Y-TgAxPP0bZ].

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

## Acknowledgements

We thank the Klymchenko lab for the kind gift of the push pull pyrene solvatochromic PK dye. We also thank Ilya Leventhal, Barbara Diaz-Roher and Volker Gerke for their critical feedback during preparation of the manuscript. In addition, we thank Matthew J Gratian and Mark Bowen of the CIMR microscopy core facility and Robin Antrobus of the CIMR proteomics facility for important technical assistance. This work was supported by the Canadian Institutes of Health Research via a Foundation Grant and Canadian Consortium on Neurodegeneration in Aging Grant (PHStGH), Alzheimer Society of Ontario (PHStGH), a Wellcome Trust Henry Wellcome Fellowship 218651/Z/19/Z (JNA), a Wellcome Trust Collaborative Award 203249/Z/16/Z (PStGH, MEV, TPK), a US Alzheimer Society Zenith Grant ZEN-18-529769 (PStGH), a National Institute on Aging grant F30AG060722 (MEW), the NIH-Oxford-Cambridge Scholars Program (MSF), the El-Hibri Foundation (MSF), the Dutch Ministry of Education – Sector Plan Beta for science and technology (FSR), an Ernest Oppenheimer Early Career Research Fellowship (TWH), the Polish Ministry of Science and Higher Education (by Mobilnosc Plus V, decision number 1623/MOB/V/2017/0, MAC), and the Medical Research Council as part of UKRI (MC_UP_1201/13 to ED & MC_U105184326 to LMB NMR facility), and the Human Frontier Science Program (Career Development Award CDA00034/2017-C to ED). The CIMR microscopy core is supported by a Wellcome Trust Strategic Award 100140, and a Wellcome Trust equipment grant 093026. The Francis Crick Institute receives its core funding from Cancer Research UK (FC001029), the UK Medical Research Council (FC001029), and the Wellcome Trust (FC001029). For the purpose of open access, the authors have applied a CC BY public copyright licence to any Author Accepted Manuscript version arising.

## Author contributions

J.N.A., F.S.R., G.W., T.W.H., M.A.C., Y.S., T.P.J.K., M.V., and P.St.G.H. designed experiments. Molecular biology and protein purification was performed by J.N.A., S.Q., W.M., J.H., H.C., and S.H.W. in the PStGH lab. Microscopy based experiments were conducted by J.N.A., G.W., J.E.C., and M.S.F. Temperature cycling experiments were performed by (Joseph) J.L.W. in the lab of ED. NMR was carried out by (Jane) JLW at the LMB core facility run by S.M.V.F. Diffusional sizing, mechanical stiffness assays and the generation of phase diagrams were performed by T.W.H., T.S., G.S., M.A.C., S.Z., Y.L., and Y.S. in the TPJK lab. Spectroscopy and nanoscale experiments were undertaken by V.P. and F.S.R. in the F.S.R.,

T.P.J.K., and M.V. labs. C.K. and J.N.A. performed experiments in cells in the M.E.W. and P.St.G.H. labs. The manuscript was written by J.N.A. and P.St.G.H. and edited by all other authors. Funding was provided by J.N.A., F.S.R., S.J.M., M.E.W., E.D., M.V., T.P.J.K., and P.St.G.H.

## Competing interests

Tuomas Knowles and Peter St George-Hyslop are co-founders of TransitionBio. Jonathon Nixon-Abell and Seema Qamar are consultants in TransitionBio. TransitionBio has no involvement in the work described in this paper, but has an interest in biomolecular condensates in cancer and infectious disease. The remaining authors declare no competing interests.

## Additional information

[1]Department of Clinical Neurosciences, Cambridge Institute for Medical Research, Clinical School, University of Cambridge, Cambridge, UK. [2]Physical Chemistry and Soft matter, Wageningen University & Research, Stippeneng, The Netherlands. [3]Yusuf Hamied Department of Chemistry, Centre for Misfolding Diseases, University of Cambridge, Cambridge, UK. [4]School of Chemical and Biomolecular Engineering, The University of Sydney, Sydney, NSW, Australia. [5]The University of Sydney Nano Institute, The University of Sydney, Sydney, NSW, Australia. [6]National institute for Neurological Disorder and Stroke, NIH, Bethesda, MD, USA. [7]Medical Scientist Training Program, Feinberg School of Medicine, Northwestern University, Chicago, IL, USA. [8]Cell Biology Division, MRC Laboratory of Molecular Biology, Cambridge, UK. [9]Structure Studies Division, NMR Facility, MRC Laboratory of Molecular Biology, Francis Crick Avenue, Cambridge Biomedical Campus, Cambridge, UK. [10]Department of Medicine (Division of Neurology), Temerty Faculty of Medicine, University Health Network, University of Toronto, Toronto, ON, Canada. [11]Carol and Gene Ludwig Center for Research on Neurodegeneration, Taub Institute for Research on Alzheimer's Disease and the Aging Brain, Department of Neurology, Columbia University Irvine Medical Center, New York, NY, USA. [12]Present address: School of Medicine, Zhejiang University, Hangzhou, China. ✉e-mail: jjn36@cam.ac.uk; ps2764@cumc.columbia.edu

