## [Transparent Peer Review File · Nature Communications]

ANXA11 biomolecular condensates facilitate protein-lipid phase coupling on lysosomal membranes

Corresponding Author: Professor Peter St George-Hyslop

Version 0:

Reviewer comments:

Reviewer #1

(Remarks to the Author)

The authors of 'ANXA11 biomolecular condensates facilitate protein-lipid phase coupling on lysosomal membranes' present a phenomenon where phase transitions in proteins and lipids might be co-regulated. Using ANXA11 as a model, the authors show that annexin recognition domain (ARD) binding to giant unilamellar vesicles (GUVs) decreases lipid disorder. The decrease in disorder was measured using techniques that include fluorescence recovery after photobleaching (FRAP) and atomic force microscopy (AFM). The authors demonstrated that ALG2 increased phase condensation of full length ANXA11, and interaction with GUV-bound ANXA11 further decreased lipid disorder. CALC decreased ANXA11 phase condensation, and interaction with GUV-bound ANXA11 increased lipid disorder. These observations were repeated in cells, where lysosomal membrane disorder was decreased in the presence of ALG2 and enhanced in the presence of CALC.

The manuscript is very well written and all concepts clearly explained. The experiments conducted were systematic and methodical, and the conclusions reached were logical. The concept of co-regulation of lipid and protein phase transitions is intriguing and a significant contribution in terms of understanding cellular organization. The following are some questions and comments to the authors:

1. For the data presented in Figure 3A and B, because PI(3)P is known to bind ANXA11, labeling the phospholipid to BODIPY- PI(3)P for use as reporter will only validate the already known PI(3)P- ANXA11 interaction, and not serve as evidence for membrane liquid-to-gel transition.
2. Is the PK dye solely sensing the lipid organization? Based on Figure 3C, a section of the dye protrudes to the lipid surface. Any protein interaction with the dye will affect its fluorescence. Is there a chance for the dye to interact with the protein (LCD/ALG2/etc) added to the ANXA11-GUV?
3. Is there any unbound ANXA11 in the experiments conducted with GUV-ANXA11 and ALG2/CALC? The authors could add excess LCD to the FL+ALG2/CALC experiments to observe driving of phase separation.
4. It is unclear in Figure 2B whether ARD phase separates or not. The blue is too dark to see clearly on a black background.

Reviewer #2

(Remarks to the Author)

Nixon-Abell et al. explore potential coupling between the ribonucleoprotein, Anxa11, which they have recently reported to form biomolecular condensates, and biophysical changes in the organization and dynamics of membrane lipids. In particular, they demonstrate that binding of Anxa11 to membrane surfaces, which is calcium dependent, slows lipid diffusion, increases membrane rigidity, and results in spectroscopic shifts that suggest changes in lipid packing. While the data are for the most part in vitro, some correlative experiments are performed on cellular lysosomes that have been artificially swelled.

While the premise of the work is interesting, the data do not demonstrate phase separation of Anxa11 on membrane

surfaces. In particular, two phases, one of which is enriched in protein and the other of which is depleted, are never observed to co-exist on membrane surfaces. This is in contrast to several recent reports of protein systems that form condensates of membranes surfaces from multiple labs: Levental, Groves, Stachowiak, Gladfelter, etc. In lieu of such direct evidence of protein condensate formation on membranes, the authors present data on changes in membrane mechanical properties. However, as detailed in comments below, these changes are expected anytime a dense layer of proteins binds to a membrane surface, regardless of whether protein phase separation/coacervation has taken place. As such, I remain unconvinced of the paper's main thesis.

Specific comments:

1. The authors state: "The phenomenon of protein-lipid phase coupling we observe within this system offers an important template to understand the numerous other examples across the cell whereby biomolecular condensates closely juxtapose cell membranes."

This phenomenon has already been shown at the immunological synapse by multiple authors: Groves, Levental. The abstract needs to be rewritten to acknowledge that the present work is extending this established idea to a new system rather than introducing a new concept.

2. The authors state: "However, direct evidence for such phase coupling has been restricted to interactions between condensates and the plasma membrane 20–23."

It is not clear why this is a conceptual limitation. Why should the phenomenon or its implications be fundamentally different at other membranes?

3. The authors state: "We observe that ALG2- and CALC-mediated changes in the ANXA11 phase state evoke a coupled phase state change in the underlying lysosomal lipid membrane."

Just because you can find a set of protein and membrane compositions/conditions in vitro where the two processes reinforce one another, how do you know that it happens in the cell? If the lipid composition there is substantially different, then protein phase separation may not be sufficient to influence lipid phase separation or vice versa.

4. It is very strange that the first figure illustrates the conclusions of the paper. The authors should leave this model until the end of the paper after the reader has been presented with the evidence to support it.

5. The authors should make clear the type and concentration of crowding agent used in the condensate studies, if any.

6. Quantification of the data in Figure 4A is needed.

7. It is rather strange in Figure 4C that FL and ARD have very similar recovery, given that the authors propose that adding LCD can slow down recovery. Shouldn't the presence of LCD within FL cause recovery of LCD to be slower than ARD? How do the authors explain this?

8. The authors state: "Importantly, this transition is dependent on the concentration of ANXA11, is reversible (Figure S1C-D), and does not elicit secondary structural changes that are associated 136 with protein aggregation (Figure S1E) 29,30."

This sentence is problematic as it contains a lot of data that is not really explained. The reader is simply expected to accept the authors' conclusions, without them first explaining the evidence. Instead, the authors should take the time to explain the data that supports these claims before making them.

9. The authors state, "Taken together, these data indicate that ANXA11 binding to lysosomal-like GUV membranes 188 causes a liquid-to-gel phase transition in the underlying lipids."

I think it is going too far to say that a liquid to gel transition has occurred. Couldn't it be that the membrane has simply increased in viscosity? The rates of recovery do not change by more than 2 fold or so. It is not very dramatic. There is no direct evidence that a gel has been formed. Also, a control is needed with a protein that does not form condensates or assemble on the membrane. It is important to see how much the recovery time of the lipids changes simply owing to the presence of a layer of unassembled proteins at the membrane surface.

10. If ANXA11 is phase separating on membrane surfaces, why do the authors not see the phase coexistence on the surface of the membrane, similar to studies by other labs (Levental, Groves, Stachowiak, Gladfelter etc.)? Considering the relatively small changes in f_{rap} associated with protein addition, the inability to observe phase separation on GUVs perhaps suggests that phase separation of the protein on membrane surfaces does not occur. Instead, the protein layer may simply become more viscous, owing to local crowding effects. If proteins are phase separating, the authors should be able to observe protein-dense and protein-dilute phases simultaneously on the surface of the same GUV, as observed by several labs for several different protein systems.

11. The authors seem to not understand that crowding the membrane surface with proteins, in the absence of phase separation, will slow down lipid diffusion dramatically. This has been shown over and over again in the literature with families of non-interacting proteins. It is simply due to volume exclusion in the 2D plane of the membrane. Absent 2 phase co-existence in the plane of the membrane, the authors should not be making claims of phase transition at the membrane

surface.

12. In the cell experiments, the authors make a highly non physiological perturbation by swelling the cellular organelles osmotically. It is hard to know whether any useful conclusions can be drawn from such a study. A super resolution microscopy approach would have been much better.

13. The authors state: "Binding of ANXA11 to GUVs resulted in a substantial >2-fold increase in the relative elastic modulus (Figure 7C), reflecting the ARD-based liquid-to-gel lipid phase transition described in Figure 2."

A dense protein layer will of course increase the rigidity of the membrane surface. But this is not evidence of phase separation.

14. I feel that many of the statements in the discussion concerning possible biological implications are far too speculative for inclusion in the manuscript, as there are no experiments presented that even begin to explore or support them.

Reviewer #3

(Remarks to the Author)

The manuscript entitled "ANXA11 biomolecular condensates facilitate protein-lipid phase coupling on lysosomal membranes" authored by Nixon-Abell et al, describes a phase coupling between protein and lipid. They focused especially on the phase interaction between ANXA11 protein and lysosome membrane. They demonstrated that ANXA11 undergoes LLPS via its N-terminal disordered region and binds to phospholipids via its C-terminal annexin repeat domain. The binding of ANXA11 proteins to giant unilamellar vesicles (GUV) shifts the phospholipid state from disordered to ordered states. They also demonstrated that ANXA11-binding proteins, ALG2 and CALC, have negative and positive effects on the phase transition of ANXA11 and phospholipids, respectively. As they mentioned in Introduction, the phase coupling between proteins and lipids is attracting a great research interest in biophysics and cell biology. The experimental systems they employed were quite reasonable and the results and conclusions were clearly described in the manuscript. On the other hand, the direct evidence of "phase coupling" was not clearly demonstrated. Especially, phase state of ANXA11 on the lipid membrane was not clearly investigated. Therefore, I recommend that the following issues should be clarified to improve and increase the impact of this study.

Major issues:

1. The most important issues to be clarified on "phase coupling" is how protein phase separation ON THE LIPID BILAYER is coupled to the phase transition of the lipids. In Figure 2, the authors demonstrated the LLPS of ANXA11 (FL and LCD) in a dispersed system (at 25-50 μM range). However, what they have to demonstrate here is the molecular state of ANXA11 "on the lipid bilayer" and not in a dispersed state. In a living cell, it binds to the lysosomal membrane (phospholipids) via ARD in a stoichiometric manner before they undergoes LLPS. This means that the LLPS assay in a dispersed system does not explain the molecular state of ANXA11 on the lysosomal membrane. Therefore, it has to be clarified how membrane-anchored ANXA11 behaves and how LCD contributes to its dynamics on the membrane. If the authors are willing to prove "phase-coupling" between protein and lipid, this is critical. It may be the case that the membrane-anchored ANXA11 just assembles via LCD during lateral diffusion (NOT phase separation), and reduces the fluidity of the lipids. Alternatively, it may undergo 2D phase transition through LCD while C-terminal end is anchored to the lipids, which is similar to the 2D phase transition of polymer brush. One possible experimental system to solve this issue will be 2D (flat) lipid bilayer instead of GUV. However, other system will also be welcome.
2. A direct evidence of phase coupling was given in Figure 4, in which they used purified LCD to induce stronger LLPS of ANXA11 on the GUV. However, I'm afraid that this system is too artificial to prove "phase coupling" in a physiological condition. In this system, ANXA11 already bound to the GUV membrane, and purified LCD was added. However, this is far from a physiological situation of the lysosomal membrane; there is no "free" LCD in the cytoplasm. It is frequently observed in vitro system that increasing amount of LCD or IDR increases the propensity of LLPS, However, it does not necessarily mean that the same event happens in a living cell. I strongly agree that "phase coupling" is an important biological mechanism. And this is why I recommend that the authors should be more careful in investigating the state of ANXA11 on the lipid bilayer. They should avoid using such non-physiological system.
3. To describe the molecular behavior of ANXA11 on the membrane, the authors should perform more quantitative analysis. Because ANXA11 is anchored to the lipid via ARD, the phase transition should depend on the protein density on the lipid (the chance of lateral interaction). For example, in Figure 3, the authors demonstrated that the addition of FL to the GUV reduced the lipid mobility. If LCD interaction contributes to this, it should show concentration dependency. I'm personally curious whether it shows linear, sigmoid or discontinuous response to the protein amount. I'm also curious about the number of ARD. Does the multivalency of ANXA11-lipid interaction affect the phase behavior? These analyses may give a hint in understanding the molecular state of ANXA11 on the membrane.
4. Related to the previous comments, it was not very clear to me how LCD contributes to the reduction of lipid mobility. In Figure 3, the authors demonstrated that the addition of FL and ARD reduced the mobility of lipid to a similar extent. This means that LCD does not contribute to the mobility reduction. In Figure 4, they added extra LCD to see the "phase coupling". However, as I mentioned, this system is non-physiological. So, I strongly recommend the authors clarify how stoichiometric anchoring of ANXA11 via ARD and self-assembly via LCD is coordinated to achieve "phase coupling".

Minor issues:

5. In AFM-IR, the authors should incorporate a negative control, in which a lipid-binding protein is used instead of FL or ARD. This will exclude the possibility that a simple protein binding to the lipid (not phase separation) could affect the IR

spectra.
6. Line 209
Figure 3A > Figure 4A

Version 1:

Reviewer comments:

Reviewer #1

(Remarks to the Author)

My comments have been addressed appropriately by the authors. They have clarified that their claim of liquid-to-gel phase transition was based on results from the PK dye and AFM-IR data, not from the binding of PI(3)P to ANXA11. Based on the cited papers, I agree that the data suggests phase transition has occurred. The authors also provided additional evidence that ANXA11 could undergo condensation on lipid surfaces, and showed that ARD-lipid interactions recruit ANXA11 to the GUV membrane. This additional data further supports the idea of protein-lipid phase coupling. The authors addressed my second concern with the figure describing PK dye and corrected it such that any future misconceptions may be avoided. I appreciate the author's response to my comment regarding the addition of LCD and driving phase separation, and I agree that this can be addressed in future work. The change from the dark blue image to a grayscale one was also greatly appreciated. I have no further questions or additional comments for the authors.

Reviewer #2

(Remarks to the Author)

The authors have addressed many of the points raised in my previous review. However, a major point remains unresolved. In every in vitro study I have read claiming protein phase separation on the surface of lipid membranes, there has always been data directly showing the co-existence of a protein-enriched phase and a protein-depleted phase coexisting on the surfaces of the same membrane (GUV, SLB, etc.). This type of evidence was presented first by Rosen and Vale in their earlier papers and has also appeared in papers by Groves, Levental, and now many others. For some reason, the authors have not performed such an experiment here. I commented on it in the first round, but I think the authors misunderstood and thought that I was asking for them to image liquid ordered and disorder lipid/membrane phases. In fact, what I want to see is phase coexistence of the protein on the surface of the same membrane. Absent this evidence, I do not find any data in the paper to definitively prove that protein phase separation is happening on the membrane. This is not a difficult experiment. The authors should be able to see this phase separation in their GUV experiments. It appears to me that the authors probably just need to lower the protein concentration. It may be the case in their current images that the entire GUV is covered by the protein-enriched phase. In other words, the membrane surface area is limiting under the chosen experimental conditions such that the protein-dilute phase does not co-exist with the protein-enriched phase. However, an alternative explanation is that the proteins under study are not capable of phase segregation on the membrane surface. The authors need to formally rule that possibility out. Demonstrating phase separation of the protein in solution in the absence of the membrane is not equivalent, because the proteins may interact differently when they are conformationally restricted by membrane binding. In my view, demonstrating phase coexistence of the protein on the membrane surface is essential for publication of this study.

Reviewer #3

(Remarks to the Author)

The manuscript entitled "ANXA11 biomolecular condensates facilitate protein-lipid phase coupling on lysosomal membranes" has been greatly improved after the revision. Most of my concerns have been clarified and the whole story is more convincing and has bigger impact. On the other hand, the following point of my concern has not yet well clarified due to some inconsistent explanations of the phase behavior of ANXA11 on the lipid bilayer. This issue should be improved before the publication.

Comment from the reviewer: 1. The most important issues to be clarified on "phase coupling" is how protein phase separation ON THE LIPID BILAYER is coupled to the phase transition of the lipids. In Figure 2, the authors demonstrated the LLPS of ANXA11 (FL and LCD) in a dispersed system (at 25-50 uM range). However, what they have to demonstrate here is the molecular state of ANXA11 "on the lipid bilayer" and not in a dispersed state. In a living cell, it binds to the lysosomal membrane (phospholipids) via ARD in a stoichiometric manner before they undergoes LLPS. This means that the LLPS assay in a dispersed system does not explain the molecular state of ANXA11 on the lysosomal membrane. Therefore, it has to be clarified how membrane-anchored ANXA11 behaves and how LCD contributes to its dynamics on the membrane. If the authors are willing to prove "phase-coupling" between protein and lipid, this is critical. It may be the case that the membrane-anchored ANXA11 just assembles via LCD during lateral diffusion (NOT phase separation), and reduces the fluidity of the lipids. Alternatively, it may undergo 2D phase transition through LCD while C-terminal end is anchored to the lipids, which is similar to the 2D phase transition of polymer brush. One possible experimental system to solve this issue will be 2D (flat) lipid bilayer instead of GUV. However, other system will also be welcome.

Response from the authors >We agree with the reviewer that this is a very important point and thank them for prompting us to perform additional experiments to better demonstrate ANXA11 condensate on lipid membranes. Protein condensate on is defined by collective intermolecular interactions. We thus reasoned that we could demonstrate ANXA11 condensate on by discriminating between (i) a simple 2-part protein-lipid interaction regime, and (ii) a 3-part protein-protein-lipid collective

interaction regime. These new data (Figure 4A) demonstrate that ANXA11-lipid interactions (as mediated through the ARD) are saturated at ~10 μ M. However, collective interactions driven by ANXA11-LCD domains, fail to reach saturation even at ~30 μ M. This indicates that ANXA11 recruitment to membranes is mediated at low concentrations by ARD-lipid interactions, then upon saturation of lipid binding sites, through LCD-LCD based interactions between independent ANXA11 molecules. This collective binding is characteristic of interactions underpinning protein condensation (Korkmazhan et al., 2021). Also, we show that phase coupling effects can be mediated through changes in the phase state of the LCD of ANXA11, and do not require the ARD per se, but can be induced by a simple NTA(Ni) linker (Figure 3G-I). This experiment indicates that phase coupling can be attributed to the condensation of the LCD and is not reliant on any complicated steric interactions between the LCD and ARD domains of ANXA11 (discussed further below in response to point3).

I appreciate the new experimental results on the behavior of proteins on a lipid bilayer. I guess the figure that they mentioned in the response is not Figure 4A, but Figure 3A. This result itself is very important and partly answered my question. However, what I was concerned is the phase behavior of full-length ANXA11 on a lipid bilayer and not the simple "binding". I'm still confused which phase transition (or state) of the protein is coupled to the lipid phase state; they clearly mentioned at least three different phases of the protein and the transition between them: dispersed, condensed (liquid-like), and condensed (gel-like), but it is not clear which phase transition (from "dispersed" to "condensed (liquid-like)", or "condensed (liquid-like)" to "condensed (gel-like)") is coupled to the phase transition of the lipid (from "disordered (liquid-like)" to "ordered (gel-like)"). The authors used several different experimental systems (including some artificial systems: adding LCD fragment, Ni-NTA-lipid + His-LCD, etc.) to investigate the phase behavior of ANXA11. But I'm afraid that some descriptions on the "phase transition" or "phase states" are not clearly specified. For example, "condensed (gel-like)" state of ANXA11 can only be seen when the LCD fragment was added in such an artificial system. Then, doesn't it happen only with ANXA11? How about the condensate formed with NTA-coupled lipid bilayer and His-tagged protein? Is it condensate (liquid-like) or condensate (gel-like)? The phase diagrams that they showed in the manuscript mostly describe the boundary between dispersed and condensed phases, and do not show the boundary between "liquid-like" and "gel-like" states. This argument is also very important when we interpret the effect of ALG2 and CALC on the phase-coupling. The authors demonstrated that "*both modulators dramatically alter the dispersed-to-condensed phase boundary of ANXA11. However, they do so in opposite directions (Figure S4D). the addition of ALG2 also pushes ANXA11 condensates into a more immobile gel-like phase (Figure S4E). By contrast, CALC increases the mobility of ANXA11 condensates, driving them into a more fluid phase (Figure S4E).*". It seems that Figure S4D examined "dispersed-condensed" phase boundary, but does not explain "liquid-like to gel-like" boundary. Furthermore, the authors claimed that this phase transition by ALG2 and CLAC induced the phase transition of underlying lipids. But I wonder which phase transitions of the protein affects this lipid phase. I'm very sorry if these results are shown and explained somewhere in the manuscript, but, due to some inconsistent and insufficient descriptions on the protein phase states in different experimental systems, it was more or less hard to follow the story. So, I recommend the authors reorganize the experimental evidence of the protein phase behavior and clearly state which phase transition (or state) is coupled to the lipid phase transition (or state).

Version 2:

Reviewer comments:

Reviewer #2

(Remarks to the Author)

I apologize for the time it has taken me to review this revised manuscript. It arrived in a very busy season. I appreciate that the authors have attempted to understand and respond to my concern. However, my confusion about their argument was not resolved by their response. They argue that protein condensation on the membrane is not creating a lipid phase transition and therefore they would not expect two dimensional phase separation of the protein in the plane of the membrane. First, I want to point out that the authors are incorrect in their assertion that lipid phase separation is required for 2D phase separation of proteins on membrane surfaces. This phenomenon has been observed many times in systems with entirely uniform lipid composition - see work of Dimova, Gladfelter, Stachowiak, etc, etc. So the absence of lipid phase separation in the authors' work does not suggest that two-dimensional phase separation of proteins on membrane surfaces should be impossible. Moreover, later in the paper, the authors suggest that the protein locally orders the lipids. That seems to be a type of "lipid phase separation", which makes me expect that the protein should phase separate in the plane of the membrane all the more. I simply can't understand the authors' reasoning. They seem to say that the entire membrane surface is the substrate for condensation of protein. If so, it must mean that the saturation concentration for phase separation is reached first on the membrane surface before it is reached in solution. If that were not the case, the condensed phase would be no more likely to emerge on the membrane surface than to first emerge in solution. In that case, phase separation of the protein in the two-dimensional plane of the membrane should be observable as a transition state before saturating the membrane surface with the condensed phase and recruiting additional protein from the solution. It is likely that the authors would need to very gradually increase the concentration of protein in solution to observe this 2-D phase coexistence, as it may occur over a very limited range of concentrations, considering the protein to lipid ratio is very high in the authors' system. Nonetheless, I feel it is essential to observe this phase coexistence and examine the properties of the two phases to determine whether they are indeed liquid-like. Absent that evidence, it is not clear to me whether the authors are simply studying protein aggregation on membrane surfaces that could be an artifact of the in vitro system.

Reviewer #3

(Remarks to the Author)

The authors introduced a new terminology "dispersed", "membrane-bound", and "condensed" to distinguish different

molecular states of ANX11. It solved most of my concerns and improved the logical flow of the entire manuscript. I don't have any further comments and would like to support the publication.

Version 3:

Reviewer comments:

Reviewer #2

(Remarks to the Author)

I appreciate the authors taking the time to address my concerns. I now feel convinced that the authors are actually observing phase separation rather than aggregation of their membrane-bound protein. However, I feel it is essential that these data be included in the published manuscript. Without this data, the key premise of the paper is left open to much doubt.

Response to Reviewers' comments on Nature Communications manuscript: NCOMMS-23-27813-T Nixon-Abell et al, "ANXA11 biomolecular condensates facilitate protein-lipid phase coupling on lysosomal membranes"

Reviewer #1 (Remarks to the Author):

The authors of 'ANXA11 biomolecular condensates facilitate protein-lipid phase coupling on lysosomal membranes' present a phenomenon where phase transitions in proteins and lipids might be co-regulated. Using ANXA11 as a model, the authors show that annexin recognition domain (ARD) binding to giant unilamellar vesicles (GUVs) decreases lipid disorder. The decrease in disorder was measured using techniques that include fluorescence recovery after photobleaching (FRAP) and atomic force microscopy (AFM). The authors demonstrated that ALG2 increased phase condensation of full length ANXA11, and interaction with GUV-bound ANXA11 further decreased lipid disorder. CALC decreased ANXA11 phase condensation, and interaction with GUV-bound ANXA11 increased lipid disorder. These observations were repeated in cells, where lysosomal membrane disorder was decreased in the presence of ALG2 and enhanced in the presence of CALC.

The manuscript is very well written and all concepts clearly explained. The experiments conducted were systematic and methodical, and the conclusions reached were logical. The concept of co-regulation of lipid and protein phase transitions is intriguing and a significant contribution in terms of understanding cellular organization. The following are some questions and comments to the authors:

We thank the reviewer for their time and useful thoughts. Major text edits in the manuscript are displayed in green. Our responses to each specific comment are below:

1. For the data presented in Figure 3A and B, because PI(3)P is known to bind ANXA11, labelling the phospholipid to BODIPY- PI(3)P for use as reporter will only validate the already known PI(3)P-ANXA11 interaction, and not serve as evidence for membrane liquid-to-gel transition.

We agree with the reviewer that these data are not evidence of a liquid-to-gel membrane phase transition, and we do not use the lipid FRAP data to sustain this claim in the manuscript. These data merely show that two lipid species, PI(3)P (now Figure 2A-B) and DOPE (Figure S2A) demonstrate decreased diffusivity when bound to ANXA11. This is not particularly surprising given that ANXA11 binds to all negatively charged lipid headgroups with varying affinities (see Figure 2, panel L of Liao et al., *Cell* 2019). The PK dye and AFM-IR data throughout the rest of figure 2 provide the basis for our claim of a liquid-to-gel transition.

2. Is the PK dye solely sensing the lipid organization? Based on Figure 3C, a section of the dye protrudes to the lipid surface. Any protein interaction with the dye will affect its fluorescence. Is there a chance for the dye to interact with the protein (LCD/ALG2/etc) added to the ANXA11-GUV?

We thank the reviewer for this comment. Figure 3C (now Figure 2C) was previously misleading in drawing the dye protruding out from the lipid surface. In fact, the PK dye is extremely lipophilic and intercalates between the acyl chains of the lipids (Valanciunaite et al, *Anal Chem* 2020). We have corrected this in the revised manuscript.

3. Is there any unbound ANXA11 in the experiments conducted with GUV-ANXA11 and ALG2/CALC?

The authors could add excess LCD to the FL+ALG2/CALC experiments to observe driving of phase separation.

We have not directly quantified the proportion of unbound to bound ANXA11, but our data suggests that the reviewer is very likely to be correct that the addition of LCD would further drive the effects of protein condensation. This is certainly supported by the effects of LCD in Figure 3D, 3I and Figure S3C. We have recently developed methods to directly examine the dynamics of protein concentration changes in the dilute phase. However, we hope that the referee will agree that this very interesting question might be best addressed in a future study designed to investigate the kinetics of ANXA11 phase modulators.

4. It is unclear in Figure 2B whether ARD phase separates or not. The blue is too dark to see clearly on a black background.

We thank the reviewer for this comment. We have now changed all of the images in Figure2 (now Figure1) to greyscale for better comparison.

Reviewer #2 (Remarks to the Author):

Nixon-Abell et al. explore potential coupling between the ribonucleoprotein, Anxa11, which they have recently reported to form biomolecular condensates, and biophysical changes in the organization and dynamics of membrane lipids. In particular, they demonstrate that binding of Anxa11 to membrane surfaces, which is calcium dependent, slows lipid diffusion, increases membrane rigidity, and results in spectroscopic shifts that suggest changes in lipid packing. While the data are for the most part in vitro, some correlative experiments are performed on cellular lysosomes that have been artificially swelled.

While the premise of the work is interesting, the data do not demonstrate phase separation of Anxa11 on membrane surfaces. In particular, two phases, one of which is enriched in protein and the other of which is depleted, are never observed to co-exist on membrane surfaces. This is in contrast to several recent reports of protein systems that form condensates of membranes surfaces from multiple labs: Levental, Groves, Stachowiak, Gladfelter, etc. In lieu of such direct evidence of protein condensate formation on membranes, the authors present data on changes in membrane mechanical properties. However, as detailed in comments below, these changes are expected anytime a dense layer of proteins binds to a membrane surface, regardless of whether protein phase separation/coacervation has taken place. As such, I remain unconvinced of the paper's main thesis.

We thank the reviewer for their time and useful comments. It is important to note that the lipid composition of GUVs used within this manuscript is not predicted to undergo lipid phase separation. Instead, our data suggest a bulk phase transition of the lipids (discussed in detail in point 10 below). We agree with the reviewer that it is important to clearly demonstrate ANXA11 condensation on GUV surfaces in order to ensure that the effects we observe on lipids do not arise from surface binding alone. As such, in our revised manuscript, we have conducted key new experiments and better highlighted previously collected data to further support this point (discussed in detail in point11 below). Major text edits in the manuscript are displayed in green.

Specific comments:

1. The authors state: "The phenomenon of protein-lipid phase coupling we observe within this system offers an important template to understand the numerous other examples across the cell whereby biomolecular condensates closely juxtapose cell membranes."

This phenomenon has already been shown at the immunological synapse by multiple authors: Groves, Levental. The abstract needs to be rewritten to acknowledge that the present work is extending this established idea to a new system rather than introducing a new concept.

In our revised manuscript, we have altered the abstract to highlight this previous work and emphasise that this manuscript addresses phase coupling on an organellar membrane. We have also updated the introduction to further highlight several recent papers on phase coupling which were unpublished at the time of our initial submission.

2. The authors state: "However, direct evidence for such phase coupling has been restricted to interactions between condensates and the plasma membrane 20–23."

It is not clear why this is a conceptual limitation. Why should the phenomenon or its implications be fundamentally different at other membranes?

Our intention here is not to draw a conceptual distinction between the biophysics of phase coupling at the plasma membrane and intracellular membranes – instead, we are simply highlighting that this is, to our knowledge, the first documented case of this phenomena occurring on an organellar membrane. This is significant because plasma membrane and organellar lipid/protein compositions and membrane curvature/topology vary dramatically. Our work, and those of the other authors cited by the reviewer, indicate that these factors can influence phase coupling. Also, lysosomes and the plasma membrane have vastly different functional roles in cells, and thus understanding specific consequences of phase coupling on each of these membranes is important (as highlighted in Figure 6). To address the reviewer's concern, we have reworded the text here to clarify this point.

3. The authors state: “We observe that ALG2- and CALC-mediated changes in the ANXA11 phase state evoke a coupled phase state change in the underlying lysosomal lipid membrane.”

Just because you can find a set of protein and membrane compositions/conditions *in vitro* where the two processes reinforce one another, how do you know that it happens in the cell? If the lipid composition there is substantially different, then protein phase separation may not be sufficient to influence lipid phase separation or vice versa.

We fully agree with the referee that corroboration of *in vitro* findings with experiments in cells is important. In Figure 5 and Figure S6, we provide data from live cells that support our *in vitro* experiments (discussed further below in response to point 12). Given the small size and dynamic nature of lysosomes, performing more sophisticated experiments in cells is simply not technically feasible with current imaging approaches. However, we hope that the referee will agree that our approach is reasonable until future advances in microscopy enable more detailed dissection of phase coupling mechanisms on organellar membranes in live cells.

4. It is very strange that the first figure illustrates the conclusions of the paper. The authors should leave this model until the end of the paper after the reader has been presented with the evidence to support it.

We have now moved this figure to the end of the paper.

5. The authors should make clear the type and concentration of crowding agent used in the condensate studies, if any.

No crowding agents are used in this manuscript. ANXA11 undergoes condensation in a temperature-dependent (Figure S1 C-D) and concentration-dependent (Figure 1B) manner.

6. Quantification of the data in Figure 4A is needed.

We have now included quantification for the images displayed in Figure 4A (now) . These data are also directly supported by the phase diagrams plotted in Figure S3B.

7. It is rather strange in Figure 4C that FL and ARD have very similar recovery, given that the authors propose that adding LCD can slow down recovery. Shouldn't the presence of LCD within FL cause recovery of LCD to be slower than ARD? How do the authors explain this?

This observation is explained by the concentration of ANXA11 on the surface of GUVs being too low to undergo condensation under these experimental conditions. We have now performed additional experiments supporting this which are discussed below in response to point 10.

8. The authors state: "Importantly, this transition is dependent on the concentration of ANXA11, is reversible (Figure S1C-D), and does not elicit secondary structural changes that are associated with protein aggregation (Figure S1E) 29,30.

This sentence is problematic as it contains a lot of data that is not really explained. The reader is simply expected to accept the authors' conclusions, without them first explaining the evidence. Instead, the authors should take the time to explain the data that supports these claims before making them.

We thank the reviewer for pointing out that this section should be explained in more detail. We have now amended the text to discuss the findings more completely.

9. The authors state, "Taken together, these data indicate that ANXA11 binding to lysosomal-like GUV membranes 188 causes a liquid-to-gel phase transition in the underlying lipids."

I think it is going too far to say that a liquid to gel transition has occurred. Couldn't it be that the membrane has simply increased in viscosity? The rates of recovery do not change by more than 2 fold or so. It is not very dramatic. There is no direct evidence that a gel has been formed. Also, a control is needed with a protein that does not form condensates or assemble on the membrane. It is important to see how much the recovery time of the lipids changes simply owing to the presence of a layer of unassembled proteins at the membrane surface.

Our claim of a liquid-to-gel phase transition in the GUV membrane is based on our AFM-IR data. The strength of AFM-IR in determining the phase state of lipids is evidenced by our data being in excellent agreement with bulk FTIR data from the literature on similar lipid systems. Specifically, we show in Figure 2 an $\Delta 8$ cm⁻¹ shift in the C=O lipid peak. The magnitude of this shift is consistent with several previous reports whereby bulk spectroscopic measurements reveal a liquid to gel phase transition in lipids of similar composition (Mantsch and McElhaney, 1991; Nicolini et al., 2006; Tamm and Tatulian, 1997). We demonstrate in Figure 3G-I that a non condensate forming protein (an inert HaloTag), when conjugated to the lipid membrane through a NiNTA linker, has no effect on lipid ordering/phase state – suggesting that the effects we see of the ANXA11 ARD on lipid ordering are not as a result of protein crowding alone. Of note, the ARD of several other annexin family members (Lin et al., 2020; Menke et al, 2005) have also been shown to elicit a gel phase in lipid membranes upon binding.

10. If ANXA11 is phase separating on membrane surfaces, why do the authors not see the phase coexistence on the surface of the membrane, similar to studies by other labs (Levental, Groves, Stachowiak, Gladfelter etc.)? Considering the relatively small changes in frap associated with protein addition, the inability to observe phase separation on GUVs perhaps suggests that phase separation of the protein on membrane surfaces does not occur. Instead, the protein layer may simply become more viscous, owing to local crowding effects. If proteins are phase separating, the authors should be able to observe protein-dense and protein-dilute phases simultaneously on the surface of the same GUV, as observed by several labs for several different protein systems.

The reviewer raises an interesting point which also surprised us upon initial inspection of the data. However, following discussions with Ilya Levental, we concluded that the lipid composition of our GUVs is such that one would certainly **not** predict the formation of distinct Lo/Ld domains. To closely mimic lysosomal lipid compositions our GUVs (50% POPC; 20% POPE; 10% POPS; 10% SAPI; 5%

PI(3)P; 5% Cholesterol) are comprised of 95% POPX lipids. This is in contrast to the high DOPC content of the GUVs generated in the studies mentioned by the reviewer. While DOPC forms large microscopically visible Lo/Ld domains, POPX lipids do not form such visible subdomains. This is nicely illustrated in Figure 3 of Heberle et al., 2010 (shown below). The added green cross represents a 95% POPC, 5% cholesterol lipid mixture which falls well outside of the predicted Lo/Ld lipid phase separation region (marked by the black box). Given the phase behaviour of lipids is dominated by the tail groups, it is reasonable to assume that our lipid composition of 95% POPX, 5% cholesterol behaves similarly.

Further supporting this, our temperature shift assay (Figure S2B) also demonstrates no phase separation or formation of Lo/Ld domains. We illustrate this concept in the manuscript in the following sentence: “Of note, the composition of our GUVs is such that one would not predict phase *separation* of the lipids into distinct ordered and disordered domains. Instead, our findings indicate a bulk phase transition of the lipid membrane into a more ordered state.”. We have now inserted the above reference above into the manuscript. The distinction between an ANXA11 surface binding versus condensation model are discussed below in response to point 11.

11. The authors seem to not understand that crowding the membrane surface with proteins, in the absence of phase separation, will slow down lipid diffusion dramatically. This has been shown over and over again in the literature with families of non-interacting proteins. It is simply due to volume exclusion in the 2D plane of the membrane. Absent 2 phase co-existence in the plane of the membrane, the authors should not be making claims of phase transition at the membrane surface.

We agree with the reviewer that the distinction between surface-binding effects and condensation of ANXA11 on lipid membranes is a key point and certainly is something we have considered in depth. Below we re-outline the evidence, including key new experimental data, disambiguating these two possibilities:

- We show that ANXA11-FL recruitment to GUV membranes is mediated by Ca²⁺-dependent binding of the ANXA11-ARD to phospholipid head groups (Figure 1C-D).
- We show that this recruitment occurs entirely independently of condensate formation (because no LCD is required; Figure 1C-D).
- We show that binding of the ANXA11-ARD alone is able to cause a liquid-to-gel phase transition in the lipid membrane (Figure 2). By itself, this data supports a 'surface-binding' model, in which association of the ANXA11-ARD with membrane surfaces is sufficient to cause a lipid phase transition.
- However, we go on to show (Figure 3) that the magnitude of this ARD-based lipid phase transition can be modulated by altering the phase state of ANXA11 through its LCD. **It is this effect that we cite as evidence of phase coupling.**
- In response to the reviewer's concern that ANXA11 might not be undergoing condensation on the membrane surface, we performed additional experiments to explore the relative saturation concentrations of both ANXA11-FL and ANXA11-ARD on our GUV membranes (Figure 4A). These data allow us to discriminate between (i) a simple 2-part protein-lipid interaction regime, and (ii) a 3-part protein-protein-lipid collective interaction regime. These new data demonstrate that ANXA11-lipid interactions (as mediated through the ARD) are saturated at ~10uM. However, collective interactions driven by ANXA11-LCD domains, fail to reach saturation even at ~30uM. This indicates that ANXA11 recruitment to membranes is mediated at low concentrations by ARD-lipid interactions, then upon saturation of lipid binding sites, through LCD-LCD based interactions between independent ANXA11 molecules. This collective binding is characteristic of interactions underpinning protein condensation (Korkmazhan et al., 2021).
- We also subsequently show that the addition of purified ANXA11-LCD hemiprotein to ANXA11 FL causes a shift in the phase boundary of ANXA11 in favour of the condensed state (Figure 3B, Figure S3B) and transitions the protein into a more immobile phase (Figure 3D).
- We use FTIR to show that this ANXA11 phase transition lacks the IR spectral features associated with classical protein aggregation, and instead ANXA11 exhibits a secondary structural composition consistent with protein condensation (Figure S3D-E).
- We then demonstrate that these changes in ANXA11 phase state induce coupled changes in the phase state of lipids in the underlying membrane (Figure 3E-F).
- We show that these effects can be mediated through changes in the phase state of the LCD of ANXA11, and do not require the ARD per se, but can be induced by a simple NTA(Ni) linker (Figure 3G-I). This experiment also indicates that phase coupling can be attributed to the condensation of the LCD and is not reliant on any complicated steric interactions between the LCD and ARD domains of ANXA11.
- We then show that two proteins known to interact with ANXA11 LCD (ALG2 and CALC) can modulate ANXA11 phase state in solution (Figure 4A, S4D-E) and on GUV membranes (Figure 4B-E).
- These ALG2/CALC-mediated changes in the ANXA11 phase state cause a coupled change in the phase state of the underlying membrane lipids (Figure 4F-G).
- Using the same artificial NTA(Ni) linker as above, in place of the ARD, we show that ALG2 and CALC effects on LCD phase state are sufficient to cause a change in lipid phase state, independently of the ARD (Figure 4H).

Taken together, these data strongly support a model in which the surface binding effects of the ANXA11-ARD on lipid membranes can be modulated by coupled changes in the phase state of the ANXA11-LCD

12. In the cell experiments, the authors make a highly non physiological perturbation by swelling the cellular organelles osmotically. It is hard to know whether any useful conclusions can be drawn from such a study. A super resolution microscopy approach would have been much better.

In principle, a superresolution approach would indeed be useful to address this. However, while various superresolution techniques are capable of resolving lysosomes (as we have published previously Liao et al., 2019), the type of ratiometric imaging required for the PK dye is not compatible with any such systems (i.e. STED, Airyscan, SMLM, SIM). While these techniques help to overcome issues of spatial resolution, they are not capable of ratiometric imaging at the speeds required to visualise lysosomal membranes. Indeed, this is something we directly tested.

Thus, we briefly expose U2OS cells to a hypotonic buffer to both arrest lysosomal motion and induce lysosomal swelling. The conditions we employ are based off a well characterised method (King et al., 2020; Guillén-Samander et al., 2021; Speckner et al., 2021) designed to study lipid phase states. In the context of our experiments, the hypotonic treatment enables the acquisition of ratiometric images (slow) of specific regions of lysosomal membrane (small). Crucially, this assay is simply a direct biophysical interrogation of *in cellula* membranes. This assay is not an attempt to recapitulate a physiological scenario. We mention in the text: "It is important to note that exposure to hypotonic conditions might change the morphology of ANXA11 condensates and likely alters the precise lipid ordering of lysosomal membranes. As such, these results should be interpreted **comparatively** across conditions, and not as absolute measurements." Crucially, these caveats do not detract from conclusions drawn on the **relative** differences of lysosomal ordering following ALG2/CALC overexpression.

13. The authors state: "Binding of ANXA11 to GUVs resulted in a substantial >2-fold increase in the relative elastic modulus (Figure 7C), reflecting the ARD-based liquid-to-gel lipid phase transition described in Figure 2."

A dense protein layer will of course increase the rigidity of the membrane surface. But this is not evidence of phase separation.

We agree with the reviewer that surface binding of the ANXA11-ARD alone (i.e. with no LCD-based condensation) is sufficient to cause a change in lipid order (Figure 2). We discuss in response to point 10 why lipid phase **separation** is not seen with this GUV composition, and why instead our data support a bulk lipid phase **transition** model. The quote here from Figure 7C describes the effects on lipids of the ARD component of the ANXA11-FL, which we agree with the reviewer occurs independently of protein condensation (or phase separation as the reviewer describes it). The evidence for the effect of phase coupling on membrane rigidity instead arises from the data presented on ALG2 and CALC – which modulate ANXA11 condensation. Further evidence and new experimental data supporting the capacity of ANXA11 to undergo condensation on the membrane is outlined in response to point11 above.

14. I feel that many of the statements in the discussion concerning possible biological implications are far too speculative for inclusion in the manuscript, as there are no experiments presented that even begin to explore or support them.

We have ensured that the text in the discussion does not involve speculation unless very clearly stated, and have clarified that these are concepts/ideas for future work.

Reviewer #3 (Remarks to the Author):

The manuscript entitled "ANXA11 biomolecular condensates facilitate protein-lipid phase coupling on lysosomal membranes" authored by Nixon-Abell et al, describes a phase coupling between protein and lipid. They focused especially on the phase interaction between ANXA11 protein and lysosome membrane. They demonstrated that ANXA11 undergoes LLPS via its N-terminal disordered region and binds to phospholipids via its C-terminal annexin repeat domain. The binding of ANXA11 proteins to giant unilamellar vesicles (GUV) shifts the phospholipid state from disordered to ordered states. They also demonstrated that ANXA11-binding proteins, ALG2 and CALC, have negative and positive effects on the phase transition of ANXA11 and phospholipids, respectively. As they mentioned in Introduction, the phase coupling between proteins and lipids is attracting a great research interest in biophysics and cell biology. The experimental systems they employed were quite reasonable and the results and conclusions were clearly described in the manuscript. On the other hand, the direct evidence of "phase coupling" was not clearly demonstrated. Especially, phase state of ANXA11 on the lipid membrane was not clearly investigated. Therefore, I recommend that the following issues should be clarified to improve and increase the impact of this study.

We thank the reviewer for their time and useful comments. We agree with the reviewer that the manuscript would benefit greatly from additional evidence supporting ANXA11 condensation on lipid membranes. Below we detail new experiments to address this.

Major issues:

1. The most important issues to be clarified on "phase coupling" is how protein phase separation ON THE LIPID BILAYER is coupled to the phase transition of the lipids. In Figure 2, the authors demonstrated the LLPS of ANXA11 (FL and LCD) in a dispersed system (at 25-50 μM range). However, what they have to demonstrate here is the molecular state of ANXA11 "on the lipid bilayer" and not in a dispersed state. In a living cell, it binds to the lysosomal membrane (phospholipids) via ARD in a stoichiometric manner before they undergo LLPS. This means that the LLPS assay in a dispersed system does not explain the molecular state of ANXA11 on the lysosomal membrane. Therefore, it has to be clarified how membrane-anchored ANXA11 behaves and how LCD contributes to its dynamics on the membrane. If the authors are willing to prove "phase-coupling" between protein and lipid, this is critical. It may be the case that the membrane-anchored ANXA11 just assembles via LCD during lateral diffusion (NOT phase separation), and reduces the fluidity of the lipids. Alternatively, it may undergo 2D phase transition through LCD while C-terminal end is anchored to the lipids, which is similar to the 2D phase transition of polymer brush. One possible experimental system to solve this issue will be 2D (flat) lipid bilayer instead of GUV. However, other system will also be welcome.

We agree with the reviewer that this is a very important point and thank them for prompting us to perform additional experiments to better demonstrate ANXA11 condensation on lipid membranes. Protein condensation is defined by collective intermolecular interactions. We thus reasoned that we could demonstrate ANXA11 condensation by discriminating between (i) a simple 2-part protein-lipid interaction regime, and (ii) a 3-part protein-protein-lipid collective interaction regime. These new data (Figure 4A) demonstrate that ANXA11-lipid interactions (as mediated through the ARD) are saturated at $\sim 10\mu\text{M}$. However, collective interactions driven by ANXA11-LCD domains, fail to reach saturation even at $\sim 30\mu\text{M}$. This indicates that ANXA11 recruitment to membranes is mediated at low concentrations by ARD-lipid interactions, then upon saturation of lipid binding sites, through LCD-LCD based interactions between independent ANXA11 molecules. This collective binding is characteristic of interactions underpinning protein condensation (Korkmazhan et al., 2021).

Also, we show that phase coupling effects can be mediated through changes in the phase state of the LCD of ANXA11, and do not require the ARD per se, but can be induced by a simple NTA(Ni) linker (Figure 3G-I). This experiment indicates that phase coupling can be attributed to the condensation of the LCD and is not reliant on any complicated steric interactions between the LCD and ARD domains of ANXA11 (discussed further below in response to point3).

2. A direct evidence of phase coupling was given in Figure 4, in which they used purified LCD to induce stronger LLPS of ANXA11 on the GUV. However, I'm afraid that this system is too artificial to prove "phase coupling" in a physiological condition. In this system, ANXA11 already bound to the GUV membrane, and purified LCD was added. However, this is far from a physiological situation of the lysosomal membrane; there is no "free" LCD in the cytoplasm. It is frequently observed in vitro system that increasing amount of LCD or IDR increases the propensity of LLPS, However, it does not necessarily mean that the same event happens in a living cell. I strongly agree that "phase coupling" is an important biological mechanism. And this is why I recommend that the authors should be more careful in investigating the state of ANXA11 on the lipid bilayer. They should avoid using such non-physiological system.

We certainly agree that isolated LCD is very unlikely to occur physiologically. Instead, this experiment serves as a mechanistic proof of concept, designed to cleanly uncouple the contribution of ARD-surface binding from LCD-based condensation. To do this, we add LCD hemiprotein onto GUVs coated in either ANXA11-FL or ANXA11-ARD at a fixed Ca²⁺ concentration. The addition of LCD hemiprotein to FL-coated GUVs increases ANXA11 condensation (Figure 3B, S3B) and pushes ANXA11 into a less mobile state (Figure 3D). This ANXA11 phase state change causes a significant increase in lipid ordering within the GUV membrane (Figure 3E-F). Crucially, these effects are not observed when LCD hemiprotein is added to ARD-coated GUVs. These data indicate that the effects of the LCD hemiprotein on ANXA11-mobility, and on the underlying lipid phase state occur independently of ARD-based surface binding effects. Crucially, our bulk protein FTIR reveal that these effects on ANXA11 mobility and lipid order is due to LCD-driven ANXA11 condensation and not due to ANXA11 aggregation (Figure S3D-E).

Prompted by this reviewer's comments we have now added to Figure 4 binding curves for ANXA11-FL's association with GUV membranes (Figure 4A). In addition to providing further evidence for ANXA11 condensation on lipid membranes (discussed above in response to point1), these data also provide a framework for understanding how the concentration of full length ANXA11 available to lysosomal membranes might influence phase coupling physiologically.

However, the most direct physiological evidence for ANXA11-based phase coupling arises from our work on ALG2 and CALC. Here, we confirm that ALG2 and CALC modulate ANXA11 phase state in solution (Figure 4A) and its mobility on GUVs (Figure 4E). These ANXA11 phase state changes alter lipid ordering (Figure 4F-G). Importantly, ALG2 and CALC also have no effect on ARD surface binding (Figure S5C) or mobility (Figure S5D). These data strongly support the notion that ALG2 and CALC modulate lipid phase transitions through their effect on LCD-mediated condensation of ANXA11, and that this effect is observable in live cells (Figure 6).

3. To describe the molecular behavior of ANXA11 on the membrane, the authors should perform more quantitative analysis. Because ANXA11 is anchored to the lipid via ARD, the phase transition should depends on the protein density on the lipid (the chance of lateral interaction). For example, in Figure 3, the authors demonstrated that the addition of FL to the GUV reduced the lipid mobility. If

LCD interaction contributes to this, it should show concentration dependency. I'm personally curious whether it shows linear, sigmoid or discontinuous response to the protein amount. I'm also curious about the number of ARD. Does the multivalency of ANXA11-lipid interaction affect the phase behavior? These analyses may give a hint in understanding the molecular state of ANXA11 on the membrane.

We refer the reviewer here to our new data which explores concentration dependency of ANXA11-FL versus ANXA11-ARD -based lipid interactions (Figure 4A; discussed in detail in response to point 1). We agree that further exploring the effects of altering ANXA11 domain stoichiometry on FRAP recovery curves (and/or other downstream effects on lipids) would yield interesting and informative data. However, given that the conclusions drawn from these (very technically challenging) experiments would be interesting and additive, rather than required to sustain our existing model, we hope that the referee agrees that this work is better suited for a future study.

4. Related to the previous comments, it was not very clear to me how LCR contributes to the reduction of lipid mobility. In Figure 3, the authors demonstrated that the addition of FL and ARD reduced the mobility of lipid to a similar extent. This means that LCD does not contribute to the mobility reduction. In Figure 4, they added extra LCD to see the "phase coupling". However, as I mentioned, this system is non-physiological. So, I strongly recommend the authors clarify how stoichiometric anchoring of ANXA11 via ARD and self-assembly via LCD is coordinated to achieve "phase coupling".

The observation of similar effect sizes of ANXA11-FL and ANXA11-ARD on lipid mobility in Figure 3 (now Figure 2B) is explained by the concentration of ANXA11-FL on the surface of GUVs in this experiment being too low to undergo condensation under these experimental conditions without the addition of exogenous LCD and/or ALG2. Previously we had assumed this to be the case, but our new data in Figure 4A demonstrates this conclusively. At 1.5 μ M ANXA11-FL (the concentration used in these FRAP studies) there is no evidence of having entered the collective binding regime representative of protein condensation. We thank the reviewer for prompting us to perform these important experiments. Of note, 1.5 μ M ANXA11 was selected to best reflect physiological ANXA11 concentrations. Speculatively, this is quite interesting because it might suggest that ANXA11 exists in a more fluid state under basal conditions until either its expression/lysosomal localisation is upregulated, or it interacts with ALG2 to drive it into a more rigid state.

Minor issues:

5. In AFM-IR, the authors should incorporate a negative control, in which a lipid-binding protein is used instead of FL or ARD. This will exclude the possibility that a simple protein binding to the lipid (not phase separation) could affect the IR spectra.

AFM-IR studies are technically challenging and time consuming when conducted to extract nanometer scale molecular interactions. AFM-IR was used as key molecular technique to identify the lipid phase transition upon binding with ANXA11 FL or ARD. Then, considering the excellent agreement between the literature, AFM-IR and PK dye experiments, our controls were all performed in parallel experiments using the PK dye (e.g. Halo-Ni-NTA in Figure 3H-I).

6. Line 209

Figure 3A > Figure 4A

We thank the reviewer for spotting this mistake and it is now corrected

Reviewer #2 (Remarks to the Author):

We thank the reviewer for their time and efforts to improve our manuscript. We believe there may have been a slight misunderstanding comparing the model we propose here with prior reports of phase coupling. We outline this in response below.

The authors have addressed many of the points raised in my previous review. However, a major point remains unresolved. In every in vitro study I have read claiming protein phase separation on the surface of lipid membranes, there has always been data directly showing the co-existence of a protein-enriched phase and a protein-depleted phase coexisting on the surfaces of the same membrane (GUV, SLB, etc.). This type of evidence was presented first by Rosen and Vale in their earlier papers and has also appeared in papers by Groves, Levental, and now many others. For some reason, the authors have not performed such an experiment here. I commented on it in the first round, but I think the authors misunderstood and thought that I was asking for them to image liquid ordered and disorder lipid/membrane phases. In fact, what I want to see is phase coexistence of the protein on the surface of the same membrane. Absent this evidence, I do not find any data in the paper to definitively prove that protein phase separation is happening on the membrane. This is not a difficult experiment. The authors should be able to see this phase separation in their GUV experiments. It appears to me that the authors probably just need to lower the protein concentration. It may be the case in their current images that the entire GUV is covered by the protein-enriched phase. In other words, the membrane surface area is limiting under the chosen experimental conditions such that the protein-dilute phase does not co-exist with the protein-enriched phase. However, an alternative explanation is that the proteins under study are not capable of phase segregation on the membrane surface. The authors need to formally rule that possibility out. Demonstrating phase separation of the protein in solution in the absence of the membrane is not equivalent, because the proteins may interact differently when they are conformationally restricted by membrane binding. In my view, demonstrating phase coexistence of the protein on the membrane surface is essential for publication of this study.

The previous work referred to by the reviewer relates to the phase separation of lipids and transmembrane proteins at the plasma membrane. In such cases, the underlying lipids undergo **phase separation** into ordered and disordered domains (see the Groves paper [DOI: 10.1016/j.bpj.2020.09.017] and Levental paper [DOI: 10.1126/sciadv.adf6205]). As described in these manuscripts, this **lipid phase separation** is required for the co-existence of concentrated and dilute protein phases. We understand the referee's point. This model is well-accepted in the field and allows the clustering of membrane proteins and their subjacent signal transduction machinery that supports effective transmembrane signaling.

However, this model does not reflect our data, and does not seem to apply in the case of ANXA11 condensation on lipid membranes. Instead, our data suggests that the condensation of ANXA11 induces a **phase transition** (not phase separation) of membrane lipids across the entire surface of the membrane. Based on the reviewer's response, they seem to accept our previous argument outlining why lipid phase separation would not be expected to occur in our system. However, as noted above, without the **phase separation** of lipids, one would not predict the co-existence of a dilute and concentrated protein phase on the membrane surface. Unlike previous models on the plasma membrane, we show that ANXA11 induces a bulk transition in lipid phase state, pushing them into more ordered states. We propose that this

subverses a mechanical function, stiffening the entire ensemble in order to resist shear stress during axonal trafficking. We support this hypothesis by showing that the condensation of ANXA11 and the concomitant phase transition of the subjacent GUV membrane lipids dramatically alters the Young's modulus of the ensemble (Figure 6 above).

This model is also supported by the correlative light and electron microscopy (CLEM) images of an RNP-granule, tethered by ANXA11, covering a significant portion of the lysosomal membrane (Figure1F below from Liao et al., *Cell*, 2019).

Figure Redacted

For these reasons, unfortunately the experiments that the referee proposes will not help to address their concerns, as the experiments evoke principles that we do not believe apply here. Indeed, as we show in Figure 1C-D (below), the initial recruitment of ANXA11 to the membrane occurs in a Ca²⁺-dependent manner, entirely independently of protein condensation (the LCD domain responsible for ANXA11 condensation is entirely dispensable). This forms a thin layer of ANXA11 on GUV surfaces. Figure 3A (next page) shows that as we increase ANXA11 concentration, we drive condensation onto GUVs through cooperative interactions. Here, the dilute phase is the protein in solution, and the concentrated phase is the thin layer of ANXA11 that coats the **entirety** of the GUV, because no lipid phase **separation** occurs (simply a phase transition to a more ordered state). As a result, we do not, and we would not expect to see, co-existent dilute and concentrated protein phases on the GUV surface.

For this same reason, the experiment proposed by the reviewer of using lower ANXA11 concentrations will unfortunately not yield informative insights. Indeed, we have performed these experiments as part of Figure 1 and, in agreement with the model we outline above, we do not see the co-existence of dilute and concentrated protein phases on the GUV membranes.

We have the sense that we may not have enunciated our model clearly enough in our previous version of the manuscript. We have now modified our discussion to explicitly highlight the difference between the receptor-mediated phase coupling model, which relies on lipid phase separation, and the bulk lipid phase transition model we observe here.

We hope that the reviewer will agree that this model will provide important new insights that have not been previously widely considered by the field.

Reviewer #3 (Remarks to the Author):

The manuscript entitled “ANXA11 biomolecular condensates facilitate protein-lipid phase coupling on lysosomal membranes” has been greatly improved after the revision. Most of my concerns have been clarified and the whole story is more convincing and has bigger impact. On the other hand, the following point of my concern has not yet well clarified due to some inconsistent explanations of the phase behavior of ANXA11 on the lipid bilayer. This issue should be improved before the publication.

We thank the reviewer for their ongoing time and efforts to improve our manuscript and agree entirely that the terminology use throughout the manuscript could and should be made clearer to highlight the important distinctions between the various phase transitions occurring. We have now amended this and comment below the changes we have made.

Comment from the reviewer: 1. The most important issues to be clarified on “phase coupling” is how protein phase separation ON THE LIPID BILAYER is coupled to the phase transition of the lipids. In Figure 2, the authors demonstrated the LLPS of ANXA11 (FL and LCD) in a dispersed system (at 25-50 uM range). However, what they have to demonstrate here is the molecular state of ANXA11 “on the lipid bilayer” and not in a dispersed state. In a living cell, it binds to the lysosomal membrane (phospholipids) via ARD in a stoichiometric manner before they undergoes LLPS. This means that the LLPS assay in a dispersed system does not explain the molecular state of ANXA11 on the lysosomal membrane. Therefore, it has to be clarified how membrane-anchored ANXA11 behaves and how LCD contributes to its dynamics on the membrane. If the authors are willing to prove “phase-coupling” between protein and lipid, this is critical. It may be the case that the membrane-anchored ANXA11 just assembles via LCD during lateral diffusion (NOT phase separation), and reduces the fluidity of the lipids. Alternatively, it may undergo 2D phase transition through LCD while C-terminal end is anchored to the lipids, which is similar to the 2D phase transition of polymer brush. One possible experimental system to solve this issue will be 2D (flat) lipid bilayer instead of GUV. However, other system will also be welcome.

Response from the authors >

We agree with the reviewer that this is a very important point and thank them for prompting us to perform additional experiments to better demonstrate ANXA11 condensate on lipid membranes. Protein condensate on is defined by collective intermolecular interactions. We thus reasoned that we could demonstrate ANXA11 condensate on by discriminating between (i) a simple 2-part protein-lipid interaction regime, and (ii) a 3-part protein-protein-lipid collective interaction regime. These new data (Figure 4A) demonstrate that ANXA11-lipid interactions (as mediated through the ARD) are saturated at ~10uM. However, collective interactions driven by ANXA11-LCD domains, fail to reach saturation even at ~30uM. This indicates that ANXA11 recruitment to membranes is mediated at low concentrations by ARD-lipid interactions, then upon saturation of lipid binding sites, through LCD-LCD based interactions between independent ANXA11 molecules. This collective binding is characteristic of interactions underpinning protein condensation (Korkmazhan et al., 2021). Also, we show that phase coupling effects can be

mediated through changes in the phase state of the LCD of ANXA11, and do not require the ARD per se, but can be induced by a simple NTA(Ni) linker (Figure 3G-I). This experiment indicates that phase coupling can be attributed to the condensation of the LCD and is not reliant on any complicated steric interactions between the LCD and ARD domains of ANXA11 (discussed further below in response to point3).

I appreciate the new experimental results on the behavior of proteins on a lipid bilayer. I guess the figure that they mentioned in the response is not Figure 4A, but Figure 3A. This result itself is very important and partly answered my question. However, what I was concerned is the phase behavior of full-length ANXA11 on a lipid bilayer and not the simple “binding”. I’m still confused which phase transition (or state) of the protein is coupled to the lipid phase state; they clearly mentioned at least three different phases of the protein and the transition between them: dispersed, condensed (liquid-like), and condensed (gel-like), but it is not clear which phase transition (from “dispersed” to “condensed (liquid-like)”, or “condensed (liquid-like)” to “condensed (gel-like)”) is coupled to the phase transition of the lipid (from “disordered (liquid-like)” to “ordered (gel-like)”).

The reviewer is correct that there are several ANXA11 states discussed in the manuscript which previously may not have been clear. To avoid confusion, we have now consistently named them as follows throughout the manuscript:

- **Dispersed:** refers to non-membrane-associated soluble ANXA11
- **Membrane-bound:** refers to lipid bound ANXA11 that has **not** undergone any phase transition
- **Condensed:** refers to dispersed or membrane-bound ANXA11 that has undergone a phase transition into a concentrated phase. It should be noted that since gelation describes a spectrum of changes in the physical properties of ANXA11, we use the terms ‘more’ or ‘less’ gelled to further describe the material properties of condensed ANXA11, and no longer as a phase state *per se*.

We first demonstrate that the ARD domain is necessary and sufficient to cause ANXA11 to move from a **dispersed to membrane-bound state** (in a Ca²⁺-dependent manner). This ARD binding is sufficient to cause the membrane lipids to transition into a more ordered state (note, no *protein* phase transition occurs here). We then show that inducing **condensation** of ANXA11 (using LCD hemiprotein or ALG2) causes a further transition of the lipids into an even more ordered, gel-like state. CALC has the opposite effect.

The authors used several different experimental systems (including some artificial systems: adding LCD fragment, Ni-NTA-lipid + His-LCD, etc.) to investigate the phase behavior of ANXA11. But I’m afraid that some descriptions on the “phase transition” or “phase states” are not clearly specified. For example, “condensed (gel-like)” state of ANXA11 can only be seen when the LCD fragment was added in such an artificial system. Then, doesn’t it happen only with ANXA11?

The reviewer raises an important point that lies at the core of why we choose to use the LCD hemi protein and not FL-ANXA11 to induce the **membrane-bound to condensed** ANXA11 phase transition in our experiments.

The concentration of FL-ANXA11 (0.5uM) that we use in experiments involving lipid membranes is not sufficient to undergo **condensation** (see the lack of collective interactions at 0.5uM in Figure 3A). As the reviewer correctly states, one option then, would be to increase the concentration of FL-ANXA11 in these experiments to push beyond the condensation threshold (C_{sat}). Indeed, this effect can be observed in Figure 3A. However, as detailed above, **membrane-bound ANXA11-FL** (i.e. not condensed) is sufficient to cause the lipids to shift to a more ordered state. Thus, the addition of more FL-ANXA11 could alter the lipid membrane properties by two mechanisms – either through ARD-mediated **membrane binding**, or through LCD-mediated **condensation**. To remove the possibility of ARD-mediated effects, we use the LCD hemiprotein to drive the **membrane-bound ANXA11** into a **condensed** state and justify our argument for phase coupling.

How about the condensate formed with NTA-coupled lipid bilayer and His-tagged protein? Is it condensate (liquid-like) or condensate (gel-like)? The phase diagrams that they showed in the manuscript mostly describe the boundary between dispersed and condensed phases, and do not show the boundary between “liquid-like” and “gel-like” states.

The reviewer will note that the concentration of the LCD hemiprotein used in the Ni-NTA experiments is 25uM (above the condensation threshold (C_{sat})). Thus the LCD protein here is in a **condensed** state (using the new, above listed terminology). We hope that the new terminology we use throughout clarifies this point. The phase diagrams thus simply report on the **dispersed to condensed** phase transition of soluble ANXA11.

This argument is also very important when we interpret the effect of ALG2 and CALC on the phase-coupling. The authors demonstrated that “*both modulators dramatically alter the dispersed-to-condensed phase boundary of ANXA11. However, they do so in opposite directions (Figure S4D). the addition of ALG2 also pushes ANXA11 condensates into a more immobile gel-like phase (Figure S4E). By contrast, CALC increases the mobility of ANXA11 condensates, driving them into a more fluid phase (Figure S4E).*”. It seems that Figure S4D examined “dispersed-condensed” phase boundary, but does not explain “liquid-like to gel-like” boundary. Furthermore, the authors claimed that this phase transition by ALG2 and CLAC induced the phase transition of underlying lipids. But I wonder which phase transitions of the protein affects this lipid phase.

We agree that this is a critical point – and is a key piece of evidence supporting the notion of phase coupling. We hope that, following the reviewer’s suggestion, the new terminology used makes this clearer. ALG2 and CALC alter the **dispersed to condensed** phase transition of ANXA11 in solution (Figure S4D). On lipid membranes, this **membrane-bound to condensed** ANXA11 phase transition is evidenced by more gel-like (ALG2) or less gel-like (CALC) physical properties depending on the modulator added (Figure S4E). As discussed above, the gelation properties of the condensed ANXA11 are not a phase state *per se*, but rather are a sliding scale of biophysical properties which we use to report on ANXA11 condensation state. As such, we are describing the **membrane-bound to condensed** ANXA11 phase transition as coupling with the underlying lipid phase

state. We thank the reviewer for helping us clarify this point and have amended the manuscript accordingly.

I'm very sorry if these results are shown and explained somewhere in the manuscript, but, due to some inconsistent and insufficient descriptions on the protein phase states in different experimental systems, it was more or less hard to follow the story. So, I recommend the authors reorganize the experimental evidence of the protein phase behavior and clearly state which phase transition (or state) is coupled to the lipid phase transition (or state).

NCOMMS-23-27813B Response to Reviewer #2

We thank the reviewer for their helpful comments (in black italic bold font). Here, we respond to each comment in blue font.

I apologize for the time it has taken me to review this revised manuscript. It arrived in a very busy season. I appreciate that the authors have attempted to understand and respond to my concern. However, my confusion about their argument was not resolved by their response.

We thank the reviewer for their continued efforts to improve our manuscript and hope that our latest efforts resolve the remaining issue. We apologise also for our rather slow reply. We too have been similarly busy with grants, students etc.

They argue that protein condensation on the membrane is not creating a lipid phase transition and therefore they would not expect two dimensional phase separation of the protein in the plane of the membrane. First, I want to point out that the authors are incorrect in their assertion that lipid phase separation is required for 2D phase separation of proteins on membrane surfaces. This phenomenon has been observed many times in systems with entirely uniform lipid composition - see work of Dimova, Gladfelter, Stachowiak, etc, etc.

The reviewer is correct that there are indeed showcases where uniform membrane lipid compositions have been used. However, they all differ from the case of ANXA11. Below we highlight the relevant papers from each of the authors mentioned, and we detail key differences with ANXA11.

We also discuss an additional experiment we have conducted, inspired by these works, to demonstrate that dilute and condensed phase ANXA11 can co-exist on GUV membranes, as the reviewer requested.

Relevant manuscripts from Dimova group:
doi.org/10.1146/annurev-biophys-030722-121518
doi.org/10.1101/2024.07.15.603610
doi.org/10.1038/s41557-023-01407-7
doi.org/10.1038/s41467-023-41709-5
doi.org/10.1038/s41467-023-37955-2

In all of the Dimova manuscripts the condensates are pre-formed (i.e. the protein is already in the condensed phase) when **wetting** onto lipid membranes. This is distinct from what we see with ANXA11 where the protein is initially recruited to the lipid membrane through the ARD domain (not condensation). Only later, once the protein accumulates on the membrane (driven through LCD-LCD interactions) do we see condensation.

Relevant manuscripts from the Gladfelter and Stachowiak group:
doi.org/10.1038/s41556-022-00882-3
doi.org/10.1038/s41467-023-43332-w

In the Gladfelter and Stachowiak manuscripts, supported lipid bilayers supplemented with Ni-NTA lipids are used to recruit various His-tagged LC domains to the membrane to undergo condensation. We exploit a similar approach in Figure 3 using His-tagged ANXA11 LCD. We therefore wondered whether we could extend this approach to address the reviewer's principal concern, **i.e. to demonstrate the co-existence of a dilute and dense phase of ANXA11 on GUV membranes.**

Our rationale here was that by generating GUVs where only a small subset of lipids could engage ANXA11-LCD, we might be able to see co-existent dilute and concentrated protein phases (that are otherwise precluded by the uniform Ca²⁺-dependent binding of ANXA11-FL across the entire GUV prior to condensation).

To accomplish this we used a 2.5% DGS-NTA(Ni) lipid composition in our GUV mix and added either 25uM His-ANXA11-LCD or 25uM His-Halo (which does not undergo condensation). Given that the His-Halo does not condense, and so there are no lateral interactions between each molecule of Halo protein, one would predict that there will be no Halo subdomains (dense/dilute phases) on the GUV surface. By contrast, if condensation, and thus lateral interactions, occur between distinct molecules of membrane-bound His- ANXA11-LCD, then one would predict this would lead to the formation of ANXA11 subdomains (dense/dilute phases) (see Fig below). Of note, **not** shown in the cartoon below are soluble LCD molecules that can also co-condense with the membrane-bound LCD molecules.

Created in BioRender. Nixon-Abell, J. (2025) <https://BioRender.com/k12w843>

We attach below representative images and quantification from this experiment. To re-emphasize, the His-ANXA11-LCD forms discrete dilute and dense phases on the GUV membranes due to lateral condensation-based interactions between adjacent His-ANXA11-LCD molecules attached to the Ni-NTA lipids in the membrane. Furthermore, but not shown in the above diagram for the sake of simplicity, additional non-membrane-bound ANXA11-LCD molecules from the soluble phase could potentially also be recruited to the newly formed condensate. In contrast, the Halo control protein does not form lateral condensation-based interactions, and so is unable to generate discrete dilute and dense phases on the GUV membranes. Instead, Halo is uniformly distributed on the membrane.

So the absence of lipid phase separation in the authors' work does not suggest that two-dimensional phase separation of proteins on membrane surfaces should be impossible. Moreover, later in the paper, the authors suggest that the protein locally orders the lipids. That seems to be a type of "lipid phase separation", which makes me expect that the protein should phase separate in the plane of the membrane all the more.

We hope that the experiment described above now clarifies how two dimensional phase separation of ANXA11 can occur on the GUV surface. Of note, we do not see lipid phase separation into distinct and separate phases following lipid binding by the ANXA11-FL. Instead, as described in the manuscript, we observe a bulk phase transition of membrane lipids wherein the binding of ANXA11-FL uniformly orders all of the underlying lipids in the membrane. Again as described in the manuscript, this was evidenced biophysically by reduced FRAP of fluorescently-labelled lipids, by spectral shifts of the solvatochromic PK dye, and by atomic force microscopy-based infrared (IR) nanospectroscopy (AFM-IR).

I simply can't understand the authors' reasoning. They seem to say that the entire membrane surface is the substrate for condensation of protein. If so, it must mean that the saturation concentration for phase separation is reached first on the membrane surface before it is reached in solution. If that were not the case, the condensed phase would be no more likely to emerge on the membrane surface than to first emerge in solution.

We are in total agreement with the reviewer on this point.

In that case, phase separation of the protein in the two-dimensional plane of the membrane should be observable as a transition state before saturating the membrane surface with the condensed phase and recruiting additional protein from the solution.

We also agree with the reviewer on this point.

As the reviewer predicts, when ANXA11-membrane occupancy is not saturated (e.g. as in the -Ni-NTA showcase above, where only 2.5% of the membrane lipids can bind to ANXA11-LCD), the dilute and dense phases can co-exist in the plane of the membrane.

However, the situation is more complex where full length ANXA11 and untagged GUV lipids are concerned. Here, ANXA11 first saturates the entire membrane through ARD-lipid interactions (i.e. no condensation). This uniform coating of ANXA11 can then condense through LCD-LCD interactions with both membrane-bound ANXA11 and with ANXA11 in the dilute phase / soluble pool of ANXA11. This condensation increases as ANXA11 concentration is increased in the dilute phase / soluble pool. The probability of any single membrane bound ANXA11 molecule interacting with any soluble ANXA11 molecule is equal. Thus, where membrane occupancy is saturated, one would not predict to see the co-existence of a dilute and dense phase on the GUV surface.

It is likely that the authors would need to very gradually increase the concentration of protein in solution to observe this 2-D phase coexistence, as it may occur over a very limited range of concentrations, considering the protein to lipid ratio is very high in the authors' system. Nonetheless, I feel it is essential to observe this phase coexistence and examine the properties of the two phases to determine whether they are indeed liquid-like. Absent that evidence, it is not clear to me whether the authors are simply studying protein aggregation on membrane surfaces that could be an artifact of the in vitro system.

As discussed above, we now provide experimental evidence of the co-existence of the dilute and dense phase of ANXA11 on GUVs. Crucially this does not occur with our Halo control.

Furthermore, both our nano-IR (Figure S1+S2) and FTIR spectra of the Amide I absorption region, reveal only minor differences (<5%) in intermolecular β -sheet content between the dispersed and

condensed protein. These biophysical readouts argue very strongly against the possibility of aggregation in our system.

We hope that the new data presented above provides sufficient evidence to convince the reviewer that the process here is indeed governed by protein condensation and not aggregation.

Reviewer 2

I appreciate the authors taking the time to address my concerns. I now feel convinced that the authors are actually observing phase separation rather than aggregation of their membrane-bound protein. However, I feel it is essential that these data be included in the published manuscript. Without this data, the key premise of the paper is left open to much doubt.

We have now included this data in the manuscript as Supplementary Figure 3F and added corresponding text into the manuscript.